



# Assessing the next generation of Global Flood Models in the Central Highlands of Vietnam

Laurence Hawker[1], Jeffrey Neal[1], James Savage[2], Thomas Kirkpatrick[1], Rachel Lord[1], Yanos Zylberberg[3], Andre Groeger[4], Truong Dang Thuy[5], Sean Fox[1], Felix Agyemang[6] , Pham Khanh Nam[5]

[1]School of Geographical Sciences, University of Bristol, Bristol, BS8 1SS, UK
[2]Fathom, Bristol, BS8 1EJ, UK
[3]School of Economics, University of Bristol, Bristol, BS8 1TU, UK
[4]Universitat Autonoma de Barcelona (UAB), Bellaterra, 08193, Spain and Barcelona School of Economics (BSE), Barcelona, 08005, Spain
[5]School of Economics, University of Economics Ho Chi Minh city, Ho Chi Minh city, 700000, Vietnam
[6]Department Planning and Environmental Management, University of Manchester, Manchester, M13 9PL, UK

*Correspondence to*: Laurence Hawker (laurence.hawker@bristol.ac.uk)

**Abstract.** Flooding is an endemic global challenge with annual damages totalling billions of dollars. Impacts are felt most acutely in in low and middle-income countries, where rapid demographic change is driving increased exposure. These areas

also tend to lack high precision hazard mapping data with which to better understand or manage risk. To address this information gap a number of Global Flood Models have been developed in recent years. However, there is substantial uncertainty over the performance of these data products. Arguably the most important component of a Global Flood Model is the Digital Elevation Model (DEM), which must represent the terrain without surface artifacts such as forests and buildings. Here we develop and evaluate a next generation of global hydrodynamic flood model based on the recently released FABDEM

DEM. We evaluate the model and compare to a previous version using the MERIT DEM at three study sites in the central highlands of Vietnam using two independent validation data sets based on a household survey and remotely sensed observations of recent flooding. The global flood model based on FABDEM consistently outperformed a model based on MERIT, and the agreement between the model and remote sensing was greater than the agreement between the two validation data sets.

## 1 Introduction

Flooding is the most frequent and deadliest natural hazard, with annual flood damages totalling billions of dollars globally (Munichre, 2020). Fatalities and impacts on livelihoods from flooding are disproportionally felt in developing countries (Jongman et al., 2012; Rentschler et al., 2022). Moreover, the impacts from flooding are expected to increase in the future due to climatic changes (Arnell and Gosling, 2016; Dottori et al., 2018) and expanding populations onto floodplains (Winsemius

et al., 2016), resulting in more people becoming exposed to flooding (Hirabayashi et al., 2021; Tellman et al., 2021; Rentschler et al., 2022) and a greater economic impact (Willner et al., 2018; Bates et al., 2023). Understanding the severity of flood hazard



is crucial in informing planning decisions to protect people and assets and for emergency response. South-East Asia is particularly at risk from severe flooding, often driven by tropical cyclones (Chen et al., 2020). Using CMIP6 climate projections, Hirabayashi et al. (2021) observe an increase in flood frequency in South-East Asia and an increase in population

exposed to flooding, with the region seeing one of the highest increases in population exposure globally.

       Vietnam is one of the most flood prone countries in the world, with a long coastline, concentrated populations and assets in deltas and floodplains, and a susceptibility to heavy precipitation primarily driven by tropical cyclones (Nguyen et al., 2021). In the past 2 decades, Vietnam has the 7th highest number of recorded disasters, with 52% classed as hydrological events (Undrr, 2020). With population exposed to flooding already high in Vietnam (38% for 50 year return period with no

flood protection), and set to increase to 46% (increase of 13-27% above current exposure) (Bangalore et al., 2019), there is a pertinent need to understand flood risk in Vietnam. In addition, there is a necessity to understand the socio-economic nuances that could exasperate or negate the risk, with Bui et al. (2014) finding from the Vietnam National Living Standard Survey of 2008 that natural disasters account for a ~6.9% reduction in household income, with floods and storms being the most common source of shock.

45       In this work, we focus on the Central Highlands of Vietnam and surrounding provinces. The region consists of the five largely mountainous inland provinces of Kon Tum, Gia Lai, Dak Lak, Dak Nong, and Lam Dong. To the east there are six coastal provinces named Quang Nam, Quang Ngai, Binh Dinh, Phu Yen, Khanh Hoa and Ninh Thuan. These eleven provinces are frequently hit by tropical cyclones, causing coastal, fluvial (river) and pluvial (surface) flooding. Examples of such events include Ketsana (2009), Mirinae (2009), Podul (2013), Damrey (2017) and Molave (2020), with Mirinae causing

an estimated 122 fatalities (Digregario, 2013). Compared to other regions in Vietnam, especially the Mekong Delta (Dung et al., 2011; Apel et al., 2016; Triet et al., 2017; Triet et al., 2020) there have been a dearth of studies on flood risk in the Central Highlands, both from a physical and socio-economic perspective. A recent review of flood literature by Nguyen et al. (2021) found no studies in the Central Highland region (five inland provinces). The Central Highlands also has some of the highest levels of poverty in the country and extensive reliance on agriculture (Le et al., 2021), making this under-researched area

suitable for further investigation.

       Flood hazard is typically mapped by running computational calculations using a hydrodynamic model. These models use information about discharge, topography and river geometry to produce hazard estimates (typically water depth and extent), usually given in terms of an annual exceedance probability (AEP) (Sampson et al., 2015; Dottori et al., 2016; Trigg et al., 2016) or sometimes by simulating a past flood event (Neal et al., 2009; Wood et al., 2016; Wing et al., 2019; Wing et al., 2021;

Alemu et al., 2023). Few locations have enough data to produce high quality flood hazard estimates, with these locations limited to the reach-scale or data-rich countries (e.g. UK or USA). Furthermore, where a good coverage of hazard maps does exist the quality and age of the underlying models can vary substantially and details regarding input data and model structure often go unreported (Bates, 2022). Global Flood Models (GFMs) can produce flood hazard information where high quality models do not exist and provide a spatially consistent modelling framework that allows for direct comparisons of hazard and

risk to be made between countries and regions e.g Rentschler et al. (2022).



However, as GFM's utilise global datasets and rely on model structures and parameterisations that have been developed and tested primarily in data rich locations (Sampson et al., 2015; Bates, 2022) there is considerable uncertainty over the efficacy of the model outputs in most locations. Studies comparing first generation GFMs (here defined as those developed five to ten years ago) found substantial differences in simulated hazard and exposure from different modelling systems. For example, Aerts et al. (2020) identified a factor of four difference between GFMs with a range of resolutions and model structures, while even similar model structures with different climate forcings had a factor of two difference. Trigg et al. (2016) compared a similar set of first generation GFMs across the African continent finding that they disagree more than they agree on the location of the 1in-in-100 year floodplain and that agreement was particularly poor over deltas. Although, on large, confined floodplains model agreement could be much higher due to valley filling where the flood extent become insensitive to magnitude and process representation. Validation work undertaken by Bernhofen et al. (2018) found that agreement between historical floods over three African floodplains observed by the MODIS satellite and GFM hazard maps varied from 40% to 70% (critical success score). The sensitivity of simulated hazard and population exposure to return period was also surprisingly low (Trigg et al., 2016), with analysis of historical satellite observations suggesting that overprediction of frequent floods was particularly problematic (Hawker et al., 2020). This bias can be expected in part due to a lack of flood defences and other anthropogenic mitigations in the GFMs, however low magnitude non-valley filling floods are also more difficult to simulate and are highly sensitive to the accuracy of elevation data and how floodplain-channel interactions are simplified in the GFM (Neal et al., 2021; Devitt et al., 2023). Furthermore, many early generation models focused on km+ resolution grids that tend to inflate exposure estimates, especially for smaller floods, due to the tendency for people to live close to but not on hazardous floodplains (Smith et al., 2019). When assessing the utility of global data sets for riverine flood risk management at national scales for five countries, Bernhofen et al. (2022) still found poor agreement between the latest GFMs, and between population datasets.

These studies showed that fine resolution (substantially sub km) is necessary and that more validation work is needed to understand the performance of different GFM frameworks across a range of locations and hazard magnitudes. A key input dataset to hydrodynamic models at the resolution needed for hazard mapping is gridded information on topography (Horritt and Bates, 2002), typically in the form of a Digital Elevation Model (DEM). Hydrodynamic models use topographic information to route flood water through floodplains, with the accuracy of the flood extent and depths a function of the similarity of the topographic data to reality. In general, a finer grid spacing in elevation datasets results in more accurate flood predictions (Fewtrell et al., 2011; Saksena and Merwade, 2015). However, finer resolution hydrodynamic simulations can yield superfluous detail at a high computational cost (Savage et al., 2016), while small increases in the ratio of DEM vertical errors to floodwater depths can substantially alter flood simulations. Thus, finer resolution simulations should generally be justified by a commensurate DEM vertical accuracy. The ability of a DEM and model to correctly represent river-floodplain interactions is also a key factor impacting the accuracy of flood simulations.

To validate GFMs, comparisons are either made to other hazard models (Trigg et al., 2016; Bernhofen et al., 2018; Fleischmann et al., 2019; Aerts et al., 2020; Bernhofen et al., 2022) or remotely sensed imagery from satellite missions [e.g.





MODIS, Sentinel] (Bates et al., 2004). Remotely sensed imagery is popular due to its global coverage, adequate resolution and availability to use on many platforms. However, remotely sensed imagery has drawbacks: the acquisition time may not match a flood event, and even if it does artifacts in the imagery such as clouds and emergent objects (e.g. vegetation and buildings) will affect the quality of the image classification. Additional validation against terrestrial observations can therefore enhance confidence by providing an independent alternative assessment of model performance. Common terrestrial validation data sources used at the reach scale include wrack/water marks (Neal et al., 2009), gauging stations, surveys and reports of fatalities/financial losses (Zischg et al., 2018). However, these have had limited use for GFM validation outside some data rich countries (Bates et al., 2010). Here, we conduct a GFM validation in Vietnam, a particularly flood-prone country classified as a lower-middle income by the World Bank.

As part of the FIERCE project, we conducted a household survey of three flood-prone villages in Dak Lak Province in the Central Highlands of Vietnam. A component of this household survey asked participants to geolocate previous floods in their land parcels. Such data has not been previously used to validate GFMs, but similar data types have been used to validate local models (Rollason et al., 2018; Petersson et al., 2020). Unlike other disciplines, validation sources are rarely cross-referenced in flood model evaluation, thus with the data available here we can analyse the effectiveness of household survey data to validate GFMs and compare it to remotely sensed imagery.

The overall aim of this research are (a) to integrate the latest global DEM data into an existing GFM, (b) cross-validate this model with remote sensing imagery and household survey data, and (c) compare with previous versions and discuss the impact this has on hazard and population exposure estimates.

## 2 Hydrodynamic Modelling Materials and methods

Hydrodynamic simulations were conducted using the University of Bristol/Fathom Global Flood Model (GFM) (Sampson et al., 2015). Since this model has been described elsewhere only a brief description of the primary model components is included here with references to more detailed model descriptions. The model utilises a 1D approximation of the St Vennant equations that neglects convective acceleration terms (Bates et al., 2010) to provide rapid yet accurate simulations under gradually varied flow conditions. To help represent floodplain dynamics across a large domain, a sub-grid channel structure is used where the governing equations are arranged on a 2D gridded floodplain with a 1D river network embedded in the grid as described in the LISFLOOD-FP model (Neal et al., 2012).

All river basins with an upstream catchment area $>50km^2$ are simulated, with river network information taken from resampling MERIT-Hydro (Yamazaki et al., 2019) to 1 arc second, which is the finest DEM resolution used in this study. Model boundary conditions are generated from a regionalised flood frequency analysis calculated at a global scale (Smith et al., 2015), which links river discharge and rainfall measurements in gauged catchments to ungauged catchments based on climatological and upstream catchment characteristics. This approach is necessary for our study area as the gauge record is sparse and temporally limited. River bathymetry for the sub-grid channels is estimated from the flood frequency, DEM and



river network data as described by Neal et al. (2021). As a result, flood hazard estimates (inundation extent and water depth) are produced in terms of a return period (RP), or its inverse and annual exceedance probability (AEP).

All catchments with an upstream area of less than 50km$^2$ are represented within a pluvial flood model. The pluvial model utilises the same channel network as the fluvial model, but simulates rainfall directly onto the DEM using the same LISFLOOD-FP solver (following the method introduced by Sampson et al. (2013)). Pluvial boundary conditions are determined by the global rainfall Intensity-Duration-Frequency (IDF) relationships (methodology described by Sampson et al. (2015)). These relationships are calculated for a number of locations globally, are pooled together and split by climate classification (based on the Koppen-Geiger classifications), with Annual Average Rainfall and climate classification used as predictors in selecting the appropriate IDF relationships for the location of interest. Again, such an approach is necessary as pre-existing IDF relationships do not exist for our study locations. We simulate pluvial flooding for the 1, 6 and 24 hour extreme rainfall durations which are combined (taking the max of each layer) to produce a pluvial flood hazard estimate, in the same format as the fluvial approach.

## 2.1 Evolution of GFM DEM data and treatment of DEM data in this research

The University of Bristol/Fathom GFM has been implemented with a variety of DEMs, with the FABDEM implementation in this paper the most recent addition. For national scale implementations, such as Bates et al., 2010 in the US and Bates et al., 2023 in the UK, significant improvements in inundation accuracy over global models have been shown where LiDAR derived elevations makes up a high proportion of the DEM. The global scale DEM (GDEM) data available for GFM simulations has undergone significant development over the last two decades. Below we review the major developments in GFM relevant global DEMs before setting out the two DEM's that will be compared in this paper.

The Shuttle Radar Topography Mission (SRTM) DEM has been the topographic dataset of choice for the majority of GFMs as it provides openly accessible elevation data at a 3 arc-second grid spacing (~90m at the equator), between 56°S to 60°N and with a mean average error of ~6m (Rodriguez et al., 2006). SRTM has errors from noise and striping, and biases from vegetation and buildings that are not conducive to high quality hydrodynamic simulations that require information on the terrain of the earth as opposed to the surface. In other words, hydrodynamic models require a Digital Terrain Model (DTM) as opposed to a Digital Surface Model (DSM). The MERIT DEM (Yamazaki et al., 2017) reduced most of these aforementioned errors, except buildings, and thus became an improved SRTM derived source of 3 arc second topographic information to use in GFMs (Hawker et al., 2020). In 2020, SRTM was reprocessed to create NASADEM, at 1 arc-second resolution (~30m at the equator) (Crippen et al., 2016), however this dataset includes vegetation and buildings in the terrain models, limiting its current potential for flood hazard mapping.

The TanDEM-X mission by German Space Agency (DLR) produced a global DEM passed on multiple satellite overpasses between 2011-2015 (Rizzoli et al., 2017). A free-to-download version at 3 arc seconds grid spacing (~90m) called TanDEM-X 90 was released in 2019 and was found to be more accurate in floodplain locations than other contemporary global DEMs, except if the floodplain contained trees (Hawker et al., 2019). Subsequently, a 1 arc second (~30m) version was released



with additional processing called Copernicus GLO-30 (referred to hereafter as COPDEM30) (Airbus, 2020). Nevertheless, both TanDEM-X 90 and COPDEM30 are still DSMs. A further popular global DEM is ALOS AW3D30 DSM (hereafter ALOS) produced by JAXA (Tadono et al., 2016; Takaku et al., 2020). The ALOS DEM is at 1 arc second grid spacing, and since version 3.2, has global coverage. ALOS differs to the global DEMs mentioned here as it is produced using photogrammetry, rather than radar interferometry.

170         Numerous studies have assessed the impact of DEMs on model quality for both GFMs and bespoke local models (Archer et al., 2018; Mcclean et al., 2020). With the recent evolution of global DEMs many of these studies have reduced relevance as they do not consider the most recent versions, or exclude some DEMs entirely. Garrote (2022) found COPDEM30 had clear improvements for flood simulations over 7 other global DEMs for a case study in Mozambique, suggesting COPDEM30 is the benchmark GDEM for GFMs. Recently, FABDEM (Forest And Building removed Copernicus Digital

Elevation Model) has been released (Hawker et al., 2022b). FABDEM removes buildings and forests from the Copernicus 30m DEM (COPDEM30) (Airbus, 2020) using a Random Forest machine learning method, providing a first global terrain map at 1 arc-second grid spacing (~30m at the equator). Prior to the release of FABDEM, Guth and Geoffroy (2021) noted that the Copernicus GLO-30 DEM should become the 'gold standard' for global DEMs, thus superseding SRTM and other global DEMs. FABDEM has been found to be more accurate than both Copernicus GLO-30 and the current global DEM of

choice for GFM simulations – MERIT DEM (Hawker et al., 2022b). For instance, in floodplains, the mean error of FABDEM was -0.03m for FABDEM, and 0.17m and 0.66m for Copernicus GLO-30 and MERIT DEM respectively. FABDEM is available from [data.bris.ac.uk](data.bris.ac.uk) . Due to the infancy of FABDEM, there has been limited assessment of it's use in hydrodynamic models (Guan et al., 2023), and none using a GFM. Example GFM simulations on SRTM, MERIT and FABDEM are plotted in Fig. 1 for illustrative purposes, issues with noise in the original SRTM data can be clearly seen in Fig. 1 a) as water is unable

to propagate far over the floodplain from the channel. Therefore, in this study we compare two versions of GFMs, both based on global terrain data. These two versions are GFM MERIT (i.e. using MERIT DEM at 3 arc second grid spacing) and GFM FABDEM ((using FABDEM at 1 arc second grid spacing). We chose not to examine other GDEMs as they are nominally DSMs. Note the version of FABDEM used was FABDEM V1.0.

## 3 Model Evaluation Materials and Methods

This study uses a unique mixture of 2 validation data sets: (i) on-the-ground observations of flooding taken from a household survey, and (ii) remotely sensed imagery from the Sentinel 1 satellite.

### 3.1 House Survey Data

        In October 2019, an extensive household survey was conducted by the authors in four flood-prone villages in Dak Lak Province. In total, 947 households and approximately 4,000 individuals were surveyed on topics covering health,

education, occupation, income, consumption, transfers, credit, assets, risk/shocks. In the risks/shocks section, respondents were





asked to identify their land parcels from a cadastral map, and whether the land parcel(s) had flooded in the last 10 years. Respondents subsequently gave the month and year of the flood of their parcel. This dataset provides a unique opportunity to corroborate flood observations with remotely sensed flood observations, as well as to test the accuracy of the flood simulations from the hydrodynamic model. Additionally, the corroboration of datasets will allow us to assess the quality of the household

responses, especially in terms of recall where the quality of response decreases the more time that passes from an event [i.e. a respondent is more likely to more accurately recall a flood event from 1 year ago as opposed to one from 9 years ago] (Bell et al., 2019). For this analysis, we focussed on three villages - Buon Triet, Thon 3 Khue Ngoc Dien and Thon 6 Vu Bon. These three villages are located along the Krong Ana river in southern Dak Lak Province and experienced flooding in 2016 according to the remotely sensed record (see Fig. 2). We excluded Ea Sup village in northern Dak Lak as this village did not experience

flooding in the period of the remotely sensed record.

### 3.2 Remote Sensing Data

Sentinel 1 synthetic aperture radar imagery was processed with Google Earth Engine using a change detection approach (UNOOSA, 2021). An image taken before the flood (baseline) is compared to an image taken during/immediately after the flood (target), and a different threshold applied to classify areas in the flooded image where radar backscatter is

lowered – and thus indicating the presence of surface water. Discharge records from Giang Son (see Fig. 2. for location) suggested peak discharge occurred on 6th November 2016. A search of the catalogue of Sentinel scenes yielded no images for 6th November 2016, with this search limited to images with 'VH' polarization. However, scenes were present on the 7th, 10th and 16th of November, and were used as target images. Scenes from 4th and 12th December were also included to capture the second flood peak of 2016 that occurred in early December. Scenes from September 2016 were selected as a dry 'baseline'.

Through trial and error, a difference threshold of 1.30 was used.

### 3.3 Gauge Data

Daily discharge data for Giang Son gauge was obtained from two sources. A record from 1978 to 1992 was sourced from the Global Runoff Data Centre (GRDC), and a record from 2000 to 2020 was sourced from Economy and Environment Partnership for Southeast Asia. This gave a comparatively short record of 36 years (Fig. 3). Annual maxima discharge (AMAX)

was calculated for the record. The older data (1978-1992) only had three AMD readings over 600 $m^3/s$ (~21% of years) with an average AMAX of 396 $m^3/s$, whereas the later record (2000-2020) had eight AMAX readings (40% of years) above 600$m^3/s$, and an average of 538 $m^3/s$.

Flood frequency analysis was performed by selecting the most suitable probability distribution for the flood frequency curve. The choice of the model was assessed using three model selection criteria, namely Akaike Information Criterion (AIC),

Bayesian Information Criterion (BIC) and the Anderson-Darling goodness-of-fit test (ADC). The ADC test has been reported to perform well for small sample sizes and heavy tail distributions and performs marginally better for a sample size used in



this study [36] (Di Baldassarre et al., 2009). No model selection criterion is consistently the best, thus there is utility using multiple model selection criterion (Laio et al., 2009).

Using the nsRFA package in R (Viglione, 2020) we find the best distribution based on the AIC and BIC criteria is Pearson Type III (P3). However, the ADC test results suggest the Generalized Extreme Value (GEV) distribution is the best model, with a minimum A2 score of 0.52. As there is no clear best distribution, we follow the advice of Laio et al. (2009) and use both models for the next stage of our analysis – deriving the design flood using the Bayesian Markov Chain Monte Carlo (MCMC) method.

Reis     The Bayesian approach has the advantage over other common methods (moments, maximum likelihood) to estimate
the parameters of at-site flood frequency curves in that it can provide credibility intervals (Gaume, 2018). This is important in our analysis where we are trying to match the return period calculated from a short record to inundation maps. The Bayesian MCMC approach is comprehensively described in Reis and Stedinger (2005). We describe the Bayesian MCMC interface approach here briefly. First, initial parameter estimates are made through the posterior distribution, calculated using Bayes' theorem (Eq..1). For flood frequency, we determine the probability that a frequency function (P) has parameters θ given the
observed realizations D (i.e our data). We use this to describe the way in which one's belief about observing parameters θ for a given P are updated having observed D.

$$P(\theta|D) = P(D|\theta) \times \frac{P(\theta)}{P(D)} \tag{1}$$

P(θ|D) is the conditional probability of θ given the observed realizations (D). In other words, P(θ|D) is the posterior probability and ultimately what we would like to infer. The prior probability not taking into account any realizations (D) is
P(θ). The probability of D given θ is given by P(D|θ). The final term is P(D), which is the prior probability of D and acts as a normalizing constant. Due to the extreme difficulty in computing the normalization constant, a simulation Monte Carlo technique is used (Haddad and Rahman, 2011).

Markov Chain Monte Carlo Simulations (MCMC) sample probability distributions based on constructing a Markov Chain with the desired distribution as its equilibrium distribution (Haddad and Rahman, 2011). The algorithm used here is
the Metropolis-Hastings algorithm. The Markov chain is started from a random initial value and run for many iterations using a Gaussian proposal density. More details of the Metropolis-Hastings algorithm can be found in Metropolis et al. (1953), Hastings (1970) and Chib and Greenberg (1995).

Using the Bayesian MCMC interface, we find the return period of the 2016 flood to be ~10 years. However, when considering the 90% confidence interval the return period ranges between 6 to 44 years for the P3 distribution (Fig. 4) and 5
to 25 years for the GEV distribution (Fig. 5) for the 2016 event, thus showing the sensitivity of flood return period estimation based on the distribution used.



### 3.4 Population Data

To calculate the population exposed to flooding, we require gridded estimates of population counts. For this analysis, we chose the building constrained population estimates at 3 arc-second (~90m) from WorldPop (Stevens et al., 2015; Bondarenko et al., 2020). This version of WorldPop distributes census information from administrative units to building locations using a random forest approach with various spatial covariates. Population estimates are for 2020. The census data used to create this version of WorldPop is from the 2009 Vietnamese census but has been projected to the year 2020 and adjusted to match United Nations (UN) population estimates. An alternative method to disaggregate population is called 'unconstrained', where the population is distributed to all habitable land pixels. We avoided using such an unconstrained gridded population dataset as the unconstrained versions tend to distribute population to uninhabited land (i.e. a frequently flooded floodplain) (Stevens et al., 2019; Niveves et al., 2021), which typically leads to overestimation of flood risk (Smith et al., 2019).

### 3.5 Land Type Data

To investigate flood exposure by land use type, we used the Global Human Settlement Layer Settlement Model Grid (GHS-SMOD) (Pesaresi et al., 2019) to classify three land types – rural; peri-urban and urban. After reprojecting the data, we resampled by nearest neighbour to the resolution of each GFM. We then classified the 7 settlement typologies in GHS-SMOD to 3 classes. For our urban class, we used grid cells from the "Urban Centre", "Dense Urban Cluster" and "semi-dense urban cluster" classes in GHS-SMOD. For peri-urban, we used the "suburban or peri-urban" class in GHS-SMOD, while our rural class was made up from the remaining classes in GHS-SMOD.

### 4 GFM Evaluation

Here we evaluate the MERIT and FABDEM based GFM fluvial hazard outputs against remotely sensed flood extents and village level flood data. An analysis is also undertaken of pluvial hazard maps due to the potential for flooding on streams smaller than the minimum catchment size represented by the fluvial model. However, we first compare the remotely sensed flood extents with the household survey data and discuss the utility of these observation data.

The principal metric for this evaluation is the well-known contingency table based critical success index (CSI) that uses binary flooded or not flooded information to measure the agreement between two data sets as shown in eq.2:

$$CSI = \frac{a}{a+b+c} \tag{2}$$

In this case the spatial unit of the comparison is the land parcels reported in the household survey data, with land parcels considered flooded in the GFM simulation or remote sensing if any part of the land parcel intersects with the flood extent from these datasets. Thus $a$ is the number land parcels flooded in both data sets, and $b$ and $c$ are the number flooded land parcels uniquely flooded in one data set but not the other. Therefore, value $a$ is the number of land parcels where the data



sets agree and *b+c* equals the number of land parcels where they do not agree. The critical success index will be 1 when all land parcels are correctly simulated and 0 when none are. Note that the critical success index generally improves for rivers with larger floodplains because a greater number of land parcels can be far from the flood edge, meaning direct inter site comparison should be treated with caution.

### 4.1 Comparison of remote sensing and household survey data

Summary statistics in Table 1 indicate that Thon 3 KND has the greatest agreement between the remotely sensed flood extent from 2016 and the household survey data at 62% CSI when including land parcels reported flooded in 2016 and those reported flooded between 2009 and 2019. Land parcels only flooded in the remote sensing were twice as prevalent (101 land parcels) as land parcels only flooded in the household data (48 land parcels). The spatial arrangement of the land parcels that are dry in the household data but wet in the remote sensing is plotted in Fig. 6(a&b). The distribution of these suggest that the vast majority were likely flooded because they sit between land parcels identified as flooded in both data sets, meaning that there is likely to be a recall bias towards non reporting amongst the households. Furthermore, despite widespread remotely observed flooding in 2016, households commonly report flooding in other years that were typically observed dry in the available remote sensing data (not shown here), suggesting recall of historical flood year was also unreliable. Most of the land parcels flooded only in the household data sit close to the edge of the remotely sensed flood extent, potentially indicating that shallow flood depths are under reported in the remotely sensed data (which would be expected in locations of low flood depth due to emergent vegetation or other surface objects).

Agreement between the remotely sensed and household data at Thon 6 VB and Buon Triet was substantially lower in Table 1. In the case of Thon 6 VB a CSI score of 37% was made up of 114 land parcels where the data sets agree, but 132 land parcels that were only flooded in the household data and 61 land parcels only flooded in the remotely sensed data. As with Thon 3 KND, most of the land parcels flooded only in the remote sensing data sit in flat locations between land parcels flooded in both data sets, suggesting a recall bias towards omission of flooding in the household data (see Fig. 7(a&b)). The potential omission rates are similar between Thon 3 KND and Thon 6 VB and in both cases the recall accuracy regarding the year flooded also appears to be low.

Land parcels flooded only in the household survey data are more prevalent at Thon 6 VB than the other sites. In some cases these are topographically too high to be part of the fluvial floodplain. It is possible that some of these land parcels are flooded by pluvial events not associated with the fluvial floodplain, with 11 land parcels in the north east quarter of the village identified as flooded by households sitting within the GFM simulated pluvial floodplain but not the fluvial floodplain (pluvial flooding is discussed in more detail later in section 4.3). However, households in the villages surveyed typically own a number of land plots distributed around the village meaning it is also possible that some household land plots have been geolocated incorrectly. The remotely sensed flood extents are more fragmented at this site than the other two villages and numerous patches of flooding lack connectivity to the channel, suggesting some flooding is misclassified or unresolved at this site in the remotely sensed data or that pluvial flooding from local sources might have played a greater role here than at the other sites.





Buon Triet had the highest number of land parcels flooded only in the remotely sensed data (Table 1). As with the other villages the spatial arrangement of the land parcels (Fig. 8) suggests a substantial recall bias, although field observations indicate the presence of flood defences on the north side of the Krong Ana river potentially complicating the data at this site. If CSI is recalculated in Table 1 without penalising for flooded only in the remote sensing data (i.e. ignoring the recall bias) the metrics increase to 0.84 (Thon 3 KND) 0.46 (Thon 6 VB) and 0.74 (Buon Triet), which are scores more typical of accuracy

statistics commonly reported in the literature for SAR flood extent mapping.

In summary, both datasets appear to be useful for assessing the performance of the GFM at the three villages, however the household survey almost certainly underreports fluvial flooding, especially when the data are thinned to specify flooding in 2016. Therefore, subsequent analysis will only consider the flooded parcel count from the household survey for the combined years between 2009-2019. Consistency between the data sets in terms of hits and flooded only in the household data

is consistent with typical performance metrics for SAR derived flood extent mapping at Thon 3 and Buon Triet, with some evidence that the remote sensing under reports around the flood edge. At Thon 6 VB the remotely sensed imagery is fragmented with the household data suggesting a significant under identification of flooded areas possibly due to resolution or shallow flooding under vegetation, but this cannot be proven with the data available. In subsequent analysis we have disaggregated the analysis of hits, misses and false alarms when comparing the GFM return periods to observed data such that the observation

data error characteristics can be discussed.

## 4.2 Evaluation of GFM fluvial hazard simulations based on MERIT and FABDEM DEMs

Gauging station data at Giang Son (see Fig. 3-5) indicate that the 2016 flooding observed in the remote sensing data was most likely a 0.1 AEP (1 in 10 year) event, with confidence interval from 0.2 to 0.02 AEP due to the short record length.

The flood return period will also change in space adding additional unknown uncertainty at the villages which sit up to 20 km away from the gauge. Nevertheless, informal discussions with households and the fact that none reported flooding of their dwellings suggests that the best estimate AEP was plausible and that this was not a once in a lifetime magnitude flood.

Extreme discharge return periods in the GFM are based on regionalization of gauging station data (Smith et al., 2015) that will include significant errors at individual river reaches despite aiming to be unbiased at large scale (Devitt et al., 2021).

Uncertainties in channel conveyance and friction parameters were not assessed here and would also modulate the discharge to flood extent relationship. We therefore compare observations to simulated hazard across all AEPs for both the MERIT and FABDEM based GFM and plot CSI scores in Fig. 6 under the assumption that a better DEM will allow for more accurate simulation of the observed data at some event magnitude. As with the intercomparison of household and remotely sensed data, the unit of space used is the land parcel rather than the individual remotely sensed or GFM pixels. These locations and the

GFM simulated hazard are plotted in Figs. 7-9 subplots c&d for the MERIT GFM and subplots e&f for the FABDEM GFM.

In addition to the CSI, the proportion of observed wet and observed dry land parcels simulated as wet at each return period were plotted in Figure 10 as normalized cumulative distributions (normalization is needed due to differing numbers of





observed wet and dry cells). This visual metric avoids assessing the accuracy of a given AEP and allows the hit rate to be disaggregated from false alarms, which we believe to be necessary when interpreting the household data due to the bias outlined

above. A GFM that is inherently better able to simulate the flood hazard will see a growth in the proportion of observed wet land parcels prior to a growth in the proportion of observed dry land parcels inundated by the model (note that the metric is limited by the number of return periods simulated and the best simulation might sit between return periods). Ideally, all observed wet land parcels would be inundated before the observed dry cells (perfect model), while a model will be worse than a random guess if a greater proportion of observed dry land parcels are inundated before the observed wet land parcels.

Although, for the household survey we expect to see simulation of observed dry land parcels due to recall bias.

For all villages the peak CSI between the remote sensing and GFM simulations was greater than between the village data and GFM. For Thon 3 KND and Thon 6 VB the best model performance against both the household and remotely sensed data occurs in the range of 0.02-0.05 AEP, with very good agreement (>0.8 CSI) between FABDEM and the remotely sensed data. When disaggregated into wet and dry CDF plots the FABDEM GFM inundates almost all (80-100%) of identified wet

land parcels by 0.05 AEP at Thon 3 KND and Thon 6 VB, although inundation of dry land parcels increases around the same AEP at these villages suggesting the model optimal performance might sit just above 0.05 AEP. The increase in dry land parcels inundation at this AEP in the Thon 3 KND household survey data is nearly twice that shown with the remotely sensed data, which is likely the result of previously mentioned recall bias in the household survey data.

The MERIT GFM has a more gradual increase in the land parcels inundated suggesting a fundamentally different

inundation dynamic. For FABDEM GFM much of the valley bottom inundates at the about the same magnitude, while for the MERIT GFM the inundation extents have a relatively gradual increase with flood magnitude. MERIT DEM tends to be smoother than FABDEM due to larger filtering windows that were applied to reduce noise from the underlying SRTM DEM relative to that required when processing the Copernicus DEM data that underpins FABDEM. On the one hand this might be expected to flatten the valley floor and result in widespread inundation at a particular magnitude (and may do so on larger

floodplains), however it also reduces the definition between the valley bottom and sides at our sites such that FABDEM appears to have a clearer demarcation between valley floor and valley sides.

In summary, the FABDEM GFM was able to attain greater accuracy to both sets of observation data and simulated a sharper transition from high hazard (valley bottom) to lower hazard (valley side). Local scale inundation models based on LiDAR often show a characteristic 'valley filling' event magnitude, where a substantial area of the floodplain inundates shortly

after the river overtops its banks, followed by substantial drop in inundation extent growth with event magnitude after the river valley has filled. MERIT GFM simulated a relatively gradual increase in inundation extent with magnitude and has less fidelity at segmenting the most and least exposed locations.

At Buon Triet the MERIT GFM obtained a greater CSI, although for a very high magnitude flood around 0.005-0.002 AEP, and both GFMs best fit the data at below 0.01 AEP (> 1 in 100 year return period). Analysis of Fig. 9 indicates that many

of the location only inundated during very high magnitude flooding sit on the northern side of the village along a small tributary





that is below the minimum catchment size represented in the fluvial component of the GFM. Therefore an analysis of pluvial flood model outputs is presented next.

**4.3 Evaluation of GFM pluvial hazard simulations based on FABDEM DEM**

When simulating flood hazards, flooding caused by intense rainfall at the local scale is often simulated separately
from the fluvial flood hazard and referred to as pluvial or surface water flooding. The exact definition of this type of flooding and how it should be simulated with respect to fluvial flooding is contested. True surface water flooding where no river channels are involved in the inundation dynamics represents one definition, however all fluvial models fail to capture small streams at some scale and in practice flooding from these is often only simulated as part of a pluvial model if at all. The concept behind the modelling is further complicated by compounding effects between fluvial and surface water flooding that are rarely
simulated. There are also substantial challenges associated with capturing data for model validation at these scales due to the limited inundation extents and durations. Remote sensing often misses pluvial flooding due to its short duration and flow inundation depths, however the household survey data obtained here is useful because it observes household land parcels being impacted by flooding in a way that is insensitive to event duration, spatial extent or source and only needs to be intense enough for the household to recall the event.

In the GFM used here the demarcation between the fluvial and pluvial model occurs for any river channel with a catchment smaller than 50 km$^2$, thus the pluvial model extends far into what many practitioners will class as fluvial floodplain. Most GFMs do not include a pluvial model due to low expectations around accuracy and challenges associated with model validation, however commercially focused models, like the one used here, often include a pluvial component due to historically significant losses from such flooding (Rözer et al., 2019; Singh et al., 2023). Here we attempt one of the first evaluations of a
GFM pluvial model, but restrict the discussion of map data to FABDEM due to its greater accuracy for the fluvial modelling.

Hazard at our village sites from the FABDEM GFM pluvial flood model is mapped in Fig. 11 b,d&f, with fluvial hazard plotted in subplots a,c&e. At Thon 3 KND the pluvial hazard sits largely within the fluvial floodplain, but increases the hazard to the south and along the norther edge of the survey land parcels. The principal fluvial floodplain to the north and north-west of the village is not inundated by the pluvial model. At Thon 6 VB the pluvial model identifies increased hazard
along a topographic depression that runs from the centre towards the north-east of the village where a number of land parcels observed as wet in the household survey data only flood in the fluvial model at high magnitude. At Buon Triet the pluvial model doesn't inundate the fluvial floodplain to the south of the village, however the hazard is significantly greater along the northern edge of the village along a stream that is too small to be represented in the fluvial model.

At all three villages the CSI scores were lower for combined fluvial and pluvial hazard due to an increase in the false
alarm rate that was greater than the additional hit rate benefit. Although the inclusion of the pluvial model had less impact on the CSI scores than the choice of DEM. Cumulative density functions tracking agreement with observed wet and dry land parcels are plotted on Fig. 12. These are identical to those in Figure 10 except the flood hazard takes the maximum of the fluvial and pluvial model AEP. At Thon 3 KND and Thon 6 VB the inclusion of the pluvial model has a minor visual impact





on these plots for rare flood events (AEPs below 0.05), but somewhat increases the hit rate and false alarms for the more frequent floods. At Buon Triet the impact is more pronounced due to the tributary entering the floodplain to the north of the village as discussed previously in relation to the fluvial hazard maps.

In conclusion, the pluvial model can identify some of the hazard related to flooding from small tributaries that is not captured in the GFM fluvial model. For a risk adverse application of the GFM, considering outputs from the pluvial model is likely to be beneficial, especially for smaller rivers that sit close to the minimum catchment size represented in the fluvial

model. False alarms increase marginally faster than the hits, which might indicate the model is less accurate due to over-prediction of the hazard. However, our observations are likely to lack the fidelity to identify if the model has skill with respect to 'true' surface water flooding away from a river channel and we would therefore expect the pluvial hazard to over-predict the observation data available. In the case of the remotely sensed data, it almost certainly does not capture surface water flooding due to the expected short duration, low depth and limited spatial extent. While in the case of the household survey

data we can hypothesise, although not validate, that the under-reporting bias will likely increase for smaller scale flooding of limited spatial impact given that households are reporting flooded agricultural land rather than buildings. A different set of test cases will be needed to disaggregate the performance of the GFM with respect to surface water flooding.

## 5. Flood Exposure for the Central Highlands of Vietnam

Flood exposure (i.e. the number of people in flooded pixels in the GFM) is calculated for 11 Vietnamese provinces

for both fluvial and pluvial flooding for both versions of the GFM. Population count is taken from WorldPop data.

### 5.1 Exposure calculations

Estimates of flood exposure for the region reveal GFM FABDEM gives consistently higher exposure estimates than GFM MERIT across all return periods (Fig. 13). We compare flood exposure from GFM FABDEM by province for the 1-in-100 year flood event. The 1-in-100 year flood event is commonly used for planning purposes thus was selected. Our analysis in

Figure 14 reveals that Quang Nam province has the most exposure with 459,000 exposed to flooding, or 31.9% of the province's population. Other coastal province's of Binh Dinh, Khanh Hoa and Quang Ngai have significant exposure between 340,000 and 370,000, or 25-30% of total province population. The other coastal province in this analysis, Phu Yen, has less exposure of around 130,000, or approximately 14% of province population. Of the inland provinces, Lam Dong has the largest exposure with approximately 200,000, or ~14% of the province population. Dak Lak, the province of our villages, has exposure

of around 187,000 or 8% of the population. Quang Nam, the most exposed province has ~8x as much exposure as the least exposed (Dak Nong). Nevertheless, the exposure, and particularly the % province population exposed to flooding is significant for all regions, with even the lowest % province population exposed 7.4%.

For fluvial flooding we found 32.8% more exposure in the GFM FABDEM model compared to GFM MERIT for the 1-in-20 year flood event and 25.5% more for the 1-in-100 year flood event (~0.5 million people) (Table 2). Variation between





provinces varies significantly, with Ninh Thuan actually showing a decrease in exposure in GFM FABDEM for the 1-in-20 year flood event. Pluvial flooding gives greater exposure (Table 3) than fluvial flooding, with GFM FABDEM giving 32% and 16.1% more exposure for the 1-in-20 and 1-in-100 year flood respectively. There is even greater variation between provinces compared to fluvial flooding, with Kon Tum province having ~10% less exposure for both the 1-in20 and 1-in-100 year flood events in the GFM FABDEM model. The biggest difference is in Dak Nong province, with GFM FABDEM

estimating almost 50% more exposure compared to GFM MERIT.

Across the spectrum of return periods analysed, GFM FABDEM gives more exposure than GFM MERIT for all return periods except above the 1-in-250 year event in Kon Tum. For one of the most extreme climate scenario (SSP5-RCP8.5), Hirabayashi et al (2021) used CMIP6 data to estimate the present day 1-in-100 year flood event could become between the 1-in-20 and 1-in-50 year flood event by 2071-2100 for this region. In other words, the 1-in-100 year flood event exposure shown

in our figures could occur 2-5x more frequently than present day, significantly altering the risk profile of the region.

## 5.1 Exposure by land use type

Further analysis across land use types reveals intriguing differences between GFM MERIT and GFM FABDEM. GFM FABDEM gives greater exposure across all return periods (Fig. 17), but for all return periods has a higher proportion in urban areas. The greatest differences are at the most frequent flood return periods, where the proportion of people exposed is

approximately 7% higher in GFM FABDEM compared to GFM MERIT for the 1-in-5 year flood and the 1-in-10 year flood (Fig. 18). This could be explained by FABDEM removing buildings from the DEM, while MERIT does not. Exposure in rural areas is approximately the same % between both models. The increase in % of exposure in urban areas in the GFM FABDEM is made up with approximately the same percentage less exposure in the peri-urban areas.

Exposure by land use type varies greatly between the provinces (Fig. 19 & 20). The coastal provinces, particularly

Khanh Hoa and Ninh Thuan have a high % of exposure in urban areas, with Ninh Thuan being between 60-70% across return periods (Figure. 20). Conversely, the inland central highland provinces have a much greater composition of risk in rural areas, in particular Dak Lak and Dak Nong, with ~70% of people exposed to flooding in rural areas. With increasing urbanization throughout the region, risk profiles could shift to more urbanized areas, thus care should be taken when planning new urban developments. Flood defences were not included in any of the model simulations here and would likely make a significant

difference to the exposure, especially for FABDEM.

## 6 Conclusions

In this paper, we assess a new GFM, taking a unique approach to validate using two data sources – commonly used remote sensing data and household survey data. We find that the GFM that uses FABDEM DEM matches more closely to a relatively frequent flood event (~1-in-10 year flood event) than a GFM using the MERIT DEM. The improved grid spacing (1

arc second compared to 3 arc second), and most importantly removal of both buildings and trees, means FABDEM gives more



realistic inundation extents when compared to a GFM using MERIT, as the GFM FABDEM follows the behaviour of more local scale models built with LiDAR data that show a characteristic 'valley filling' event magnitude.

By using two sources of validation data, we found that neither are perfect to the extent that the FABDEM GFM could provide a better fit to both validation data sets (in terms of CSI) than they could to each other. Household surveys, where participants were asked to specify the year of a flood on a given land parcel, suffered from recall bias. The longer ago the flood event, the more difficulty the participants had in remembering the timing and severity of the flood. This could have been due to no 'once in a generation' flood impacting the surveyed villages in the survey period – instead there were several relatively small non-problematic floods. The remote sensing data, often assumed by practitioners to depict the 'true' flood, and thus be the most suitable data to benchmark a model, is not without its deficiencies. By using two data sources, as well as field surveys, we find that the remote sensing data is not always reliable, especially on the flood margins. Remote sensing allows us to 'see' flooding from space, but what remote sensing sees may not directly correlate to true risk. If participants do not recall events, it is unlikely the event had much impact, highlighting potential adaptations that are invisible to remotely sensed images. Therefore, when assessing flood model performance, multiple validation dataset sources are preferable and sources should be rigorously assessed. Often only remote sensing sources will be available, but wherever possible different classification schemes and/or different satellites should be used and augmented with auxiliary sources (e.g social media and media reports).

Lastly, we presented flood exposure estimates for 11 provinces in the central region of Vietnam. This helps fulfil the dearth of studies for the region (Nguyen et al., 2021) and comprehensive flood hazard information across a spectrum of flood severities. The flood hazard data presented in this paper is available from NERC EIDC (Hawker et al., 2022a). We find the coastal provinces have the most exposure, with Binh Dinh, Quang Nam and Quang Ngai having the most exposure for both fluvial and pluvial flooding. For the 1-in-100 year flood, we estimate ~2.5m will be exposed to fluvial flooding and ~3m to pluvial flooding. The proportion of people exposed to flooding in each province is significant with almost a third of the total population of Ninh Thuan and Quang Ngai exposed to the 1-in-100 year event. The FABDEM model finds a greater proportion of flood exposure in urban areas relative to MERIT DEM, prior to considering flood defences. With extreme events set to become more common, and the urbanisation in the region which can encroach onto risky floodplain areas, it is essential proper planning is implemented to minimise the flood risk.

**Acknowledgments**

This research is funded by the Vietnam National Foundation for Science and Technology Development (NAFOSTED) and Natural Environment Research Council (NERC) under grant number NE/S003061/1.



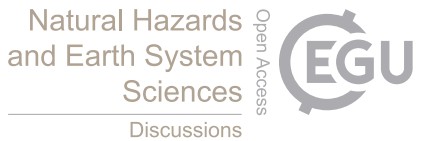
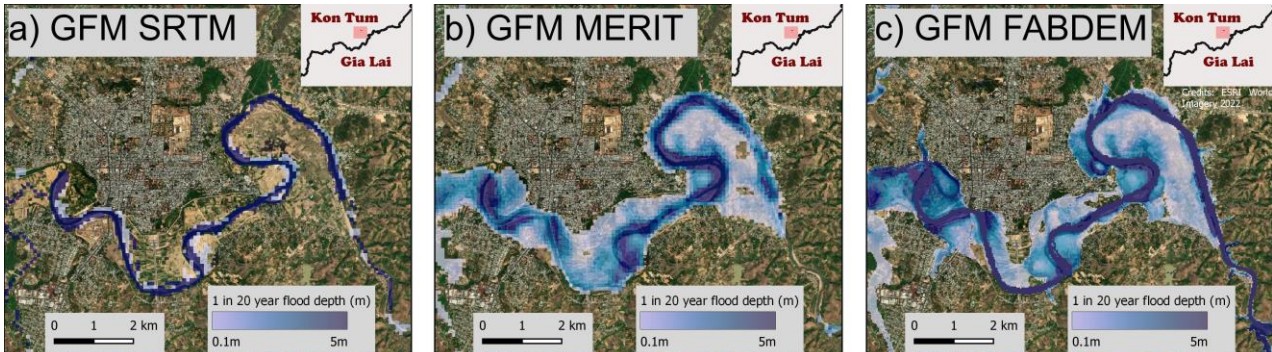

**Figure 1: Flood hazard map of Kon Tum, Vietnam, for the one in 20-year flood event. Three versions of a Global Flood Model (GFM), each with a different underlying elevation dataset, are compared. Panel a) uses the SRTM elevation dataset, Panel b) uses MERIT DEM, and Panel c) uses FABDEM. Note the improved delineation of floodplain features in panel c and lack of flood propagation in panel a. Basemap ESRI World Imagery**

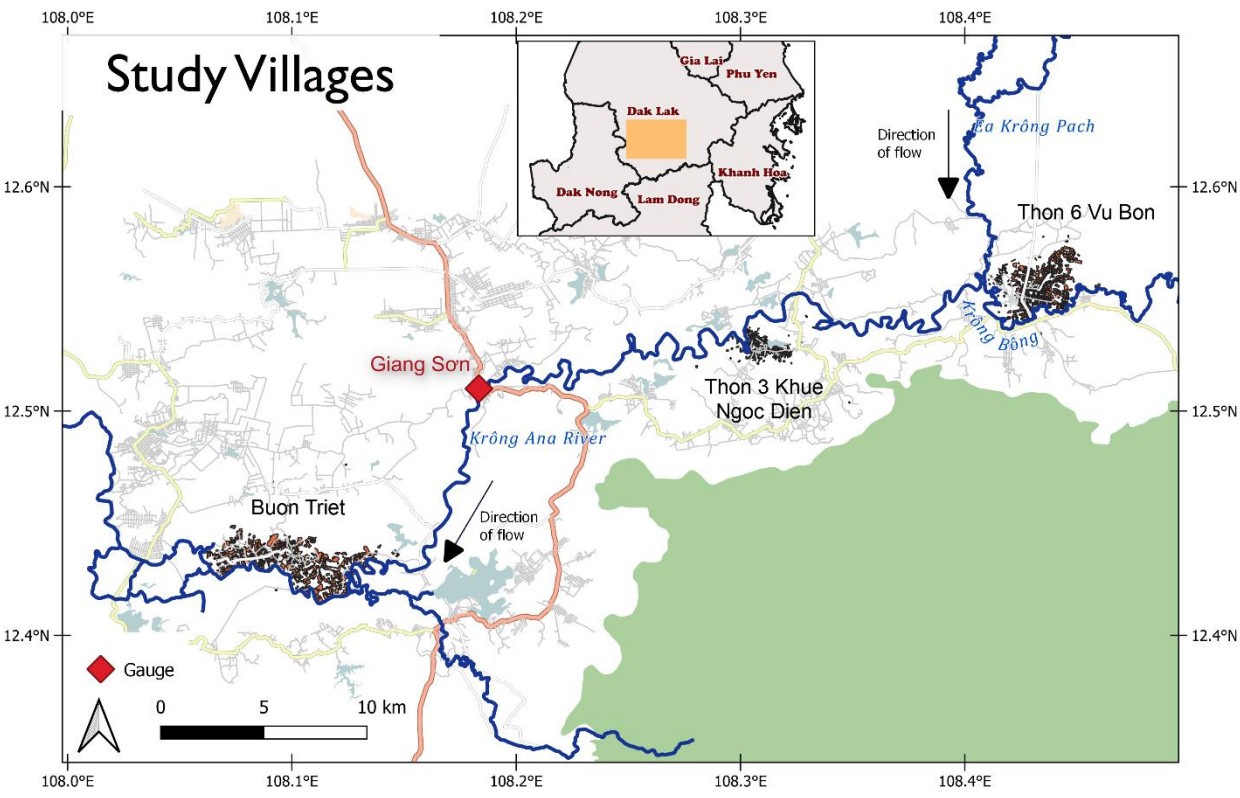

**Figure 2: Location of the three villages that were the focus of our household survey (Buon Triet, Thon 3 Khue Ngoc Dien and Thon 6 Vu Bon) in Dak Lak Province. The river gauge record is displayed with a red diamond. Basemap © OpenStreetMap contributors.**

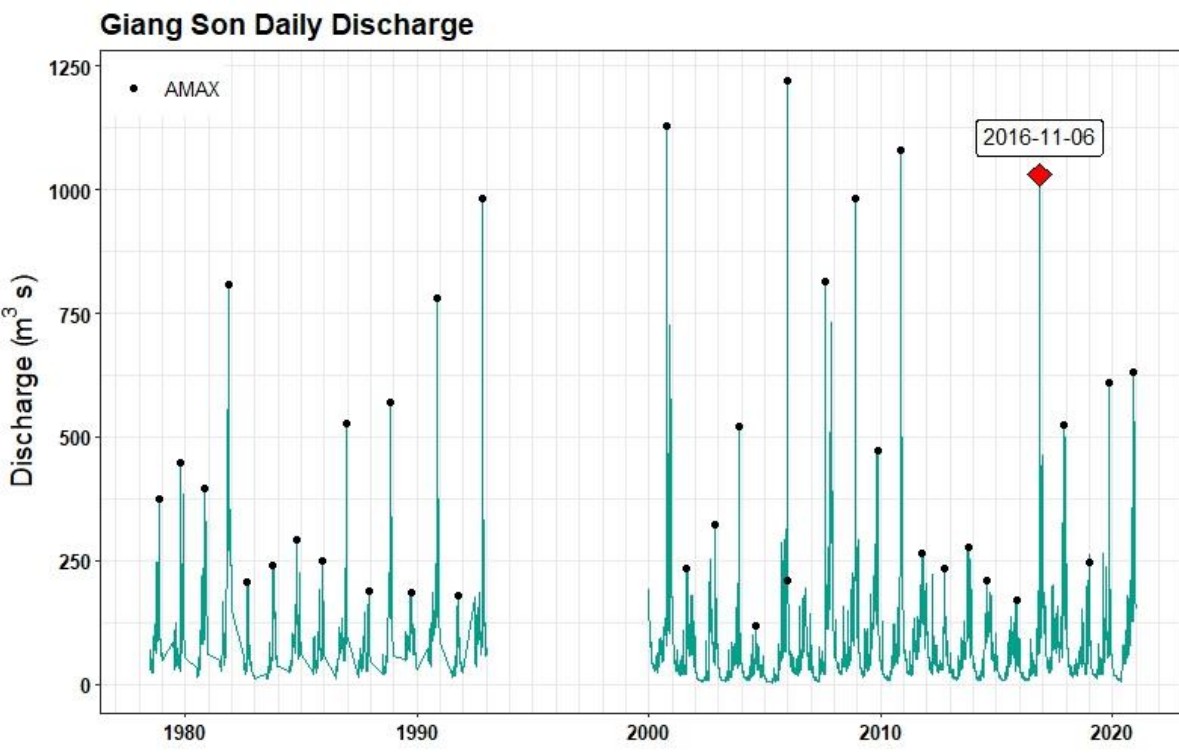

**Figure 3: Hydrograph for Giang Son Gauge. Annual flow maximum (AMAX) depicted with black dots, with the AMAX for 2016 highlighted with a red diamond.**

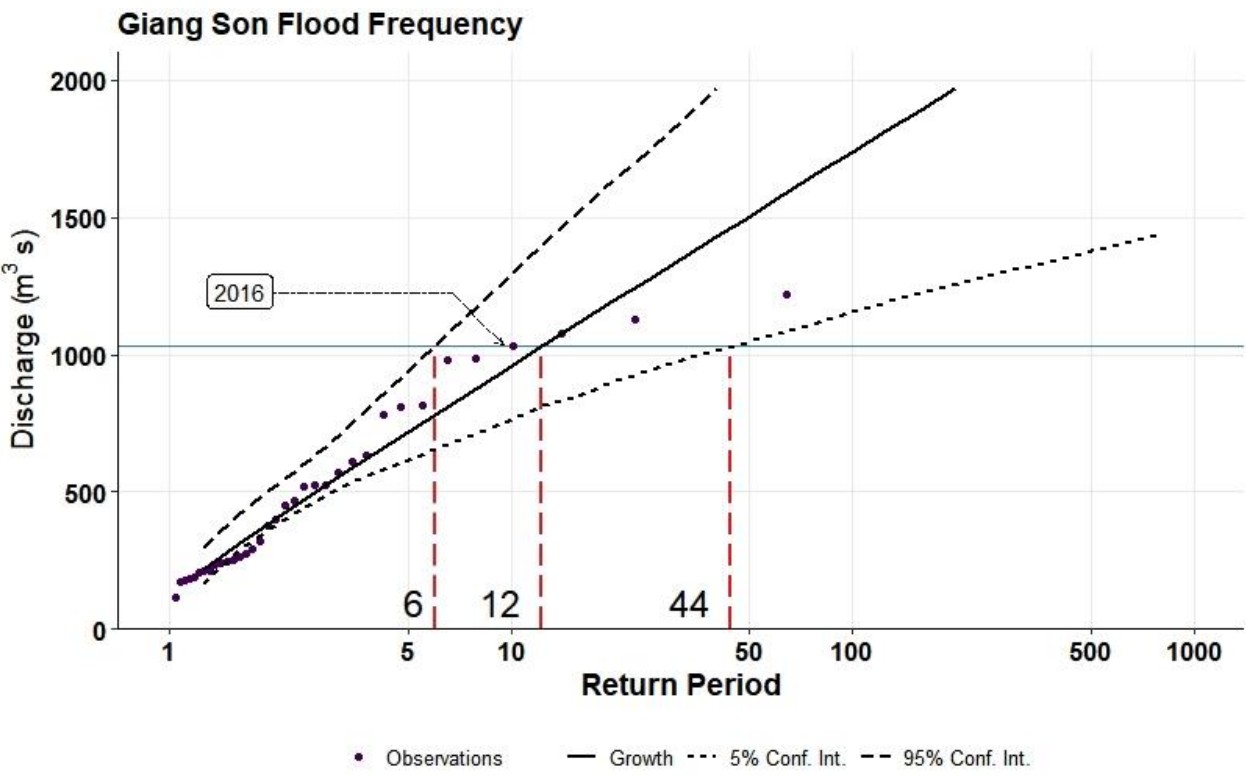

**Figure 4: Design Flood Estimation at Giang Son using the Bayesian MCMC interface and the Pearson Type III (P3) distribution.**





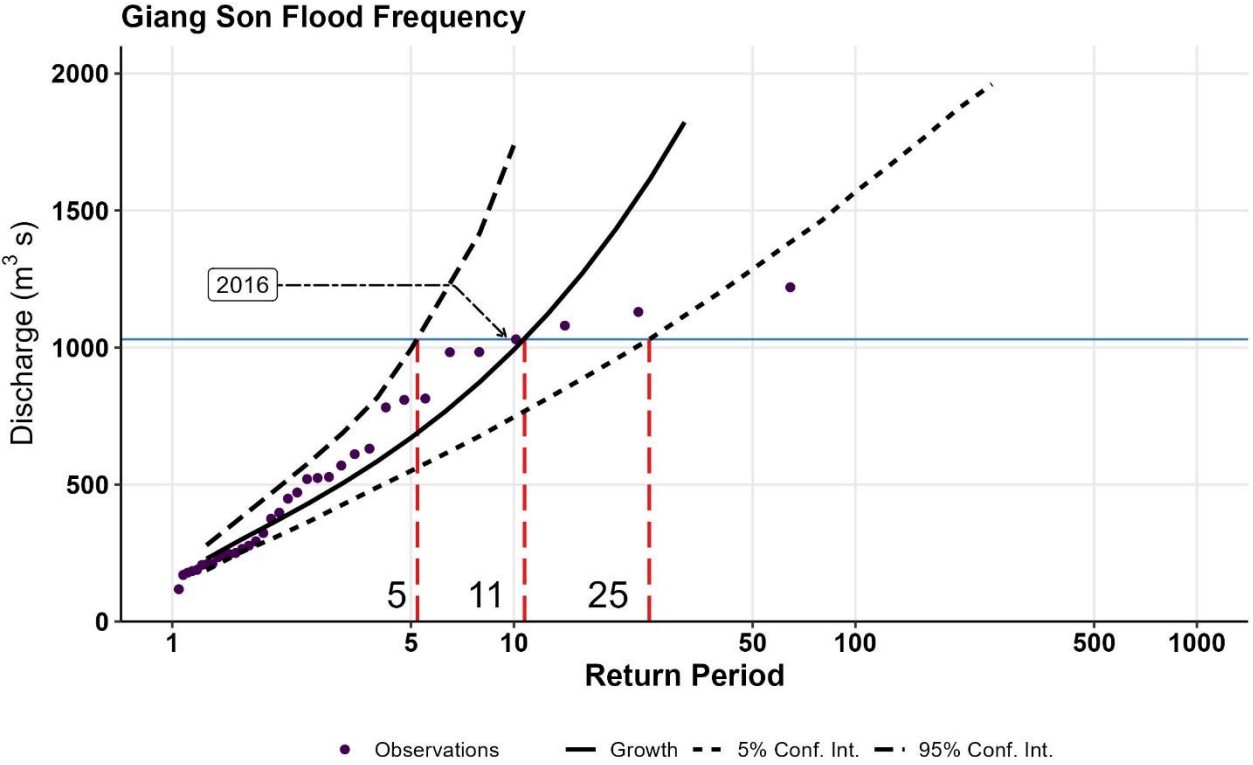

**Figure 5: Design Flood Estimation at Giang Son using the Bayesian MCMC interface and the Generalized Extreme Value (GEV) distribution.**

Table 1: Comparison of flooded land parcel data from remotely sensed imagery and household survey responses

|  | Flooded in both data sets | Flooded in only household data | Flooded only in remote sensing data | CSI (without penalising for only flooded in RS) |
|---|---|---|---|---|
| Thon 3 | 252 | 48 | 101 | 0.62 (0.84) |
| Thon 6 | 114 | 132 | 61 | 0.37 (0.46) |
| Buon Triet | 184 | 62 | 225 | 0.39 (0.74) |





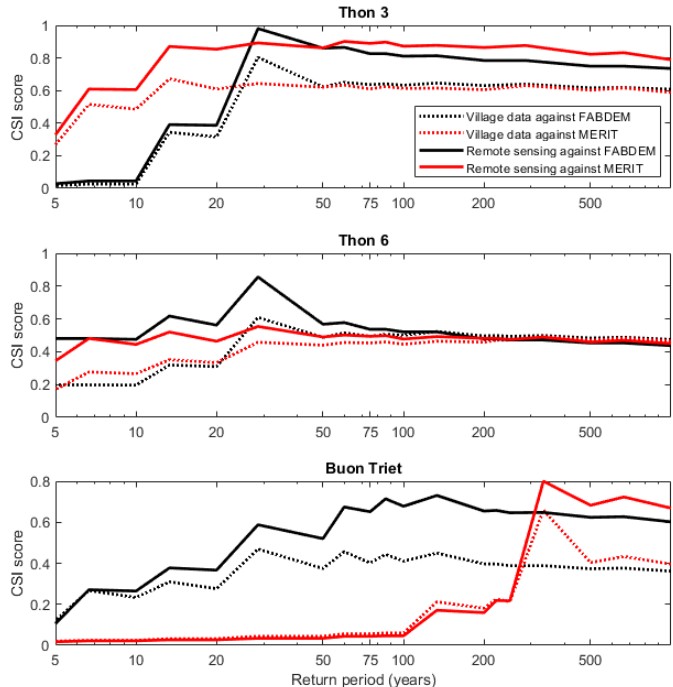

**Figure 6: Critical success index scores by return period against remotely sensed (solid lines) and household survey (dotted lines) validation data for MERIT (red) based GFM and FABDEM (black) based GFM.**




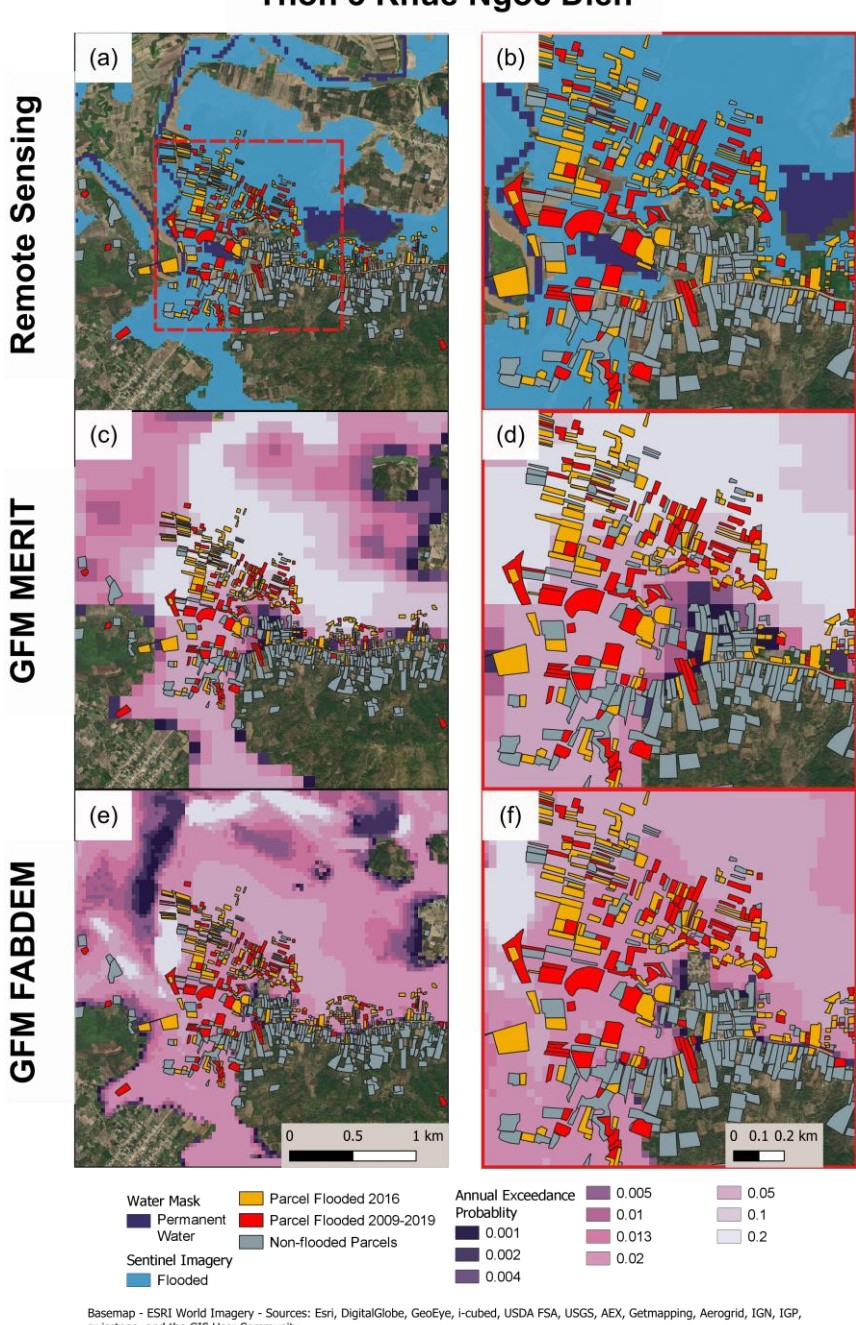

**Figure 7: Flood Maps for Thon 3 Khue Ngoc Dien. Panel (a) shows flood extent from 2016 acquired from Sentinel 1 data. Panel (c) shows annual exceedance probability (AEP) for GFM V2, and panel (e) shows AEP for GFM V3.Panels (b), (d) and (f) show zoomed in versions of the remote sensing and GFM outputs.**


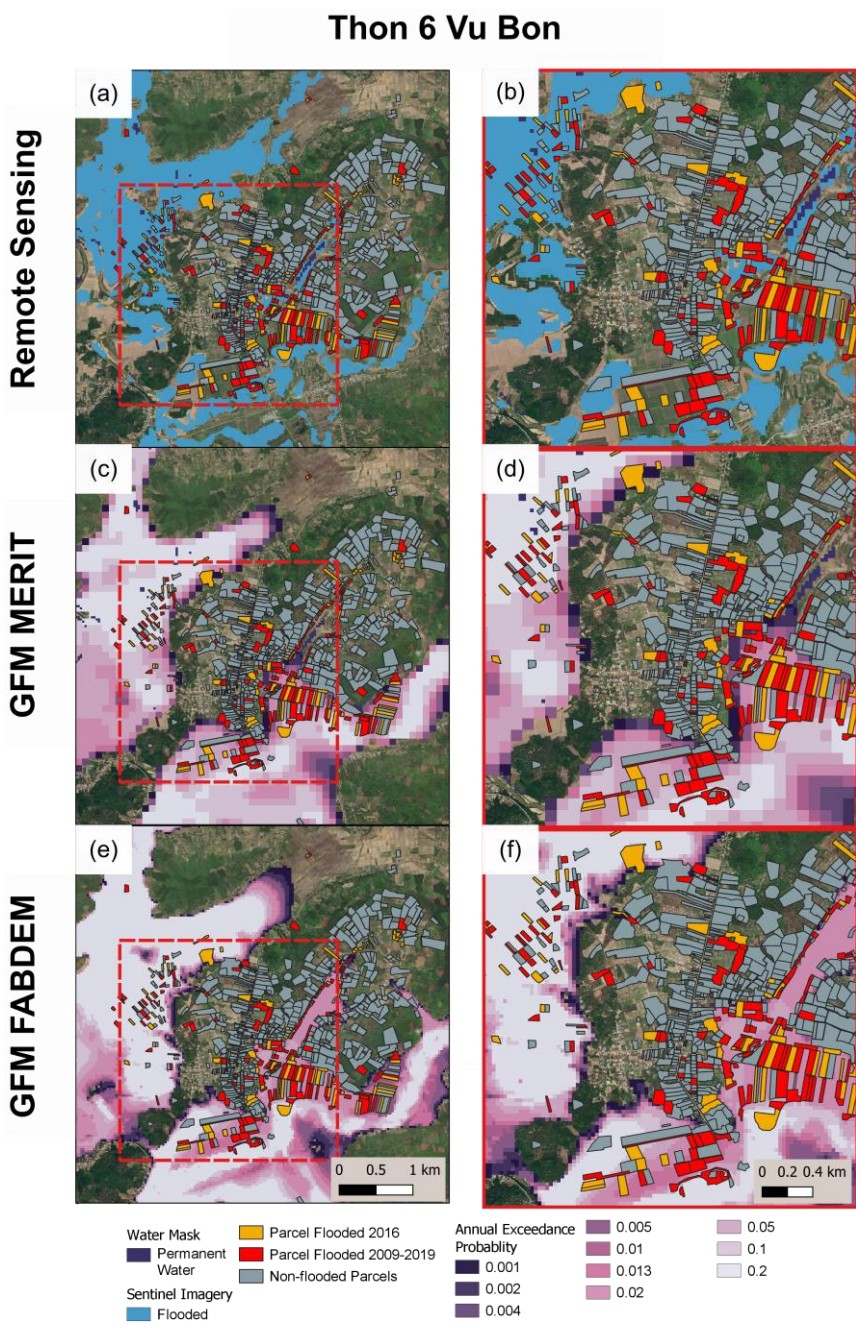

**Figure 8: Flood Maps for Thon 6 Vu Bon. Panel (a) shows flood extent from 2016 acquired from Sentinel 1 data. Panel (c) shows annual exceedance probability (AEP) for GFM V2, and panel (e) shows AEP for GFM V3.Panels (b), (d) and (f) show zoomed in versions of the remote sensing and GFM outputs.**

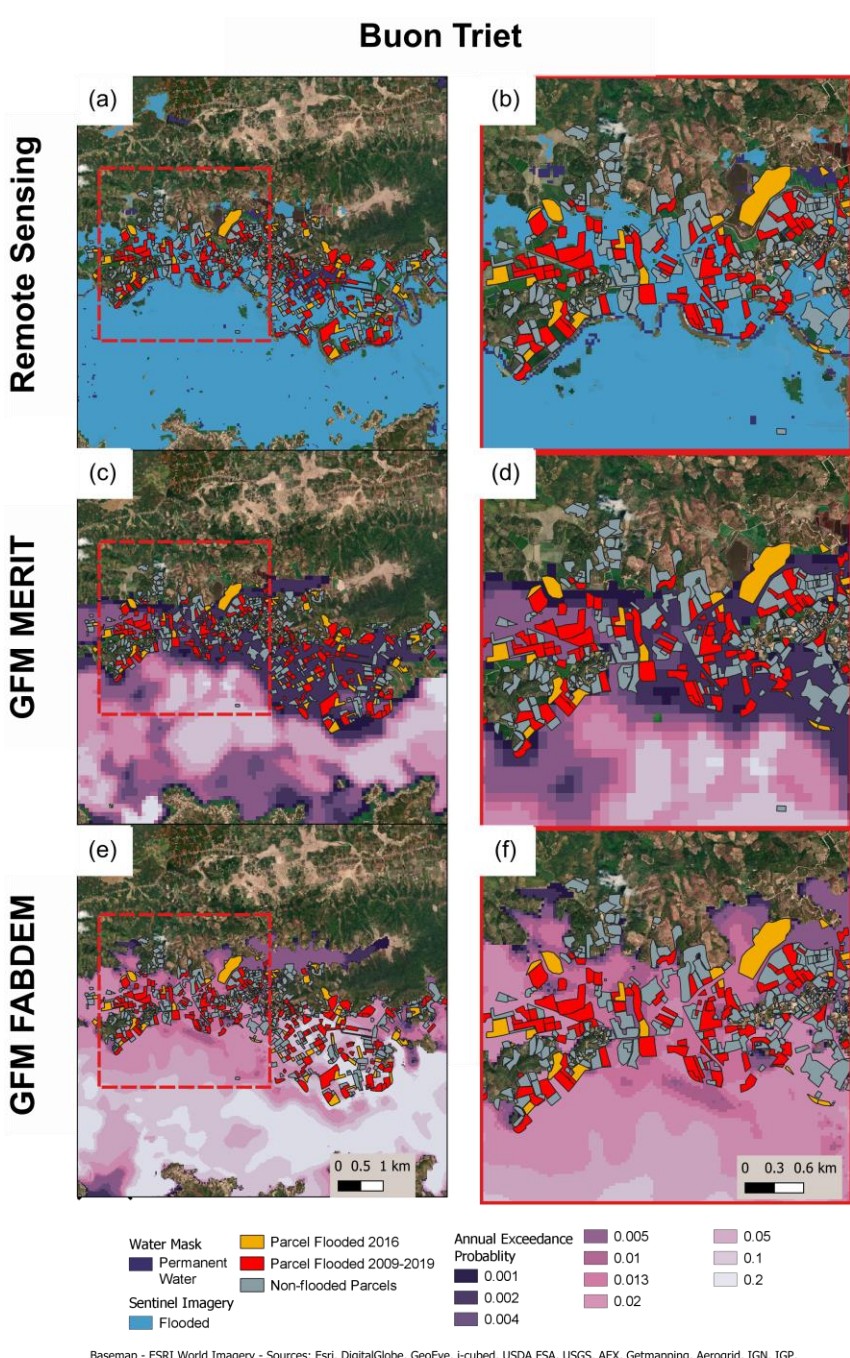

**Figure 9: Flood Maps for Thon 6 Vu Bon. Panel (a) shows flood extent from 2016 acquired from Sentinel 1 data. Panel (c) shows annual exceedance probability (AEP) for GFM V2, and panel (e) shows AEP for GFM V3. Panels (b), (d) and (f) show zoomed in versions of the remote sensing and GFM outputs.**



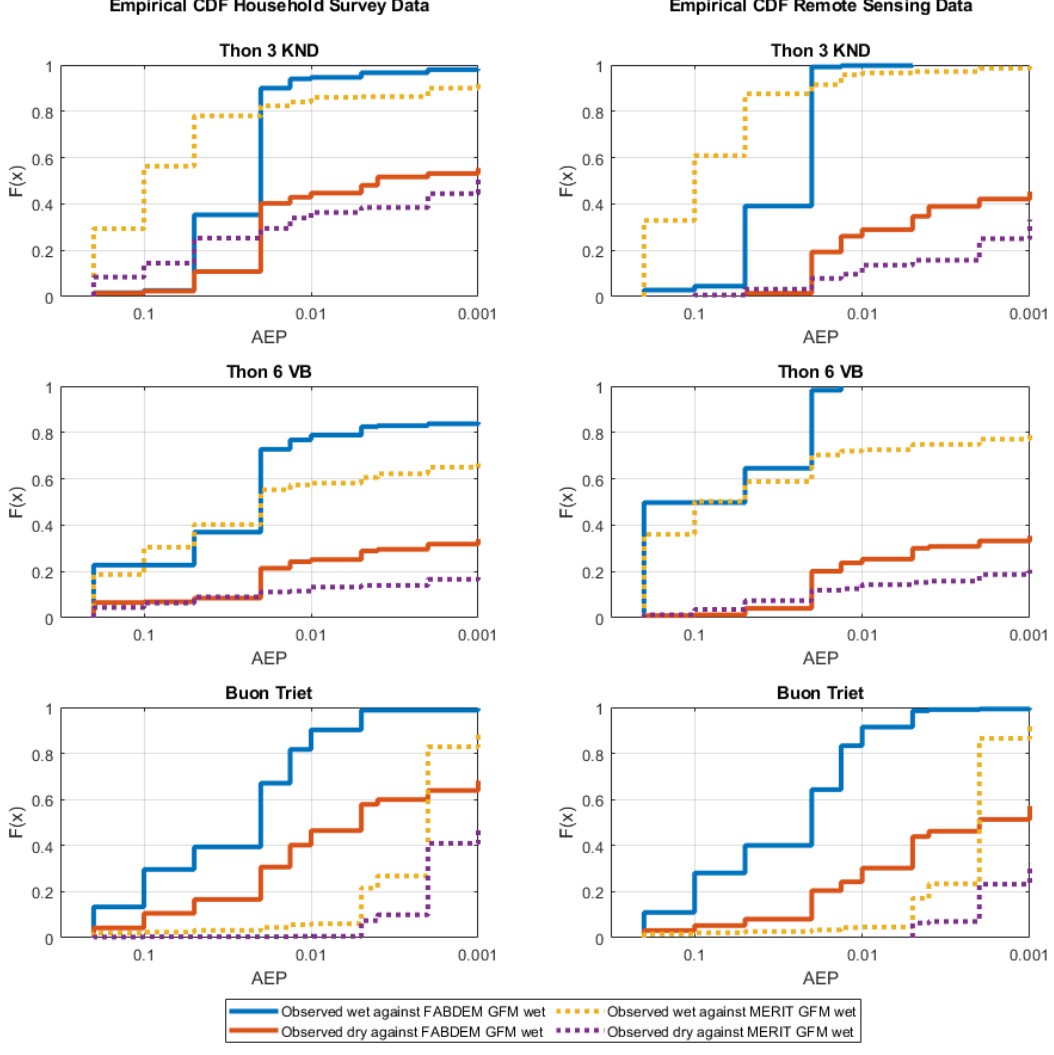

**Figure 10 : Cumulative density functions tracking agreement with observed wet and dry land parcels in the household survey data (left) and remotely sensed data (right). Solid lines are FABDEM GFM and dotted lines are MERIT GFM.**







Figure 11: Six panel fluvial vs pluvial flood maps for three villages. Only FABDEM GFM results are plotted.





**Figure 12: Cumulative density functions tracking agreement with observed wet and dry land parcels in the household survey data (left) and remotely sensed data (right). Solid lines are FABDEM GFM including combined pluvial and fluvial hazard and dotted lines are MERIT GFM including combined pluvial and fluvial.**


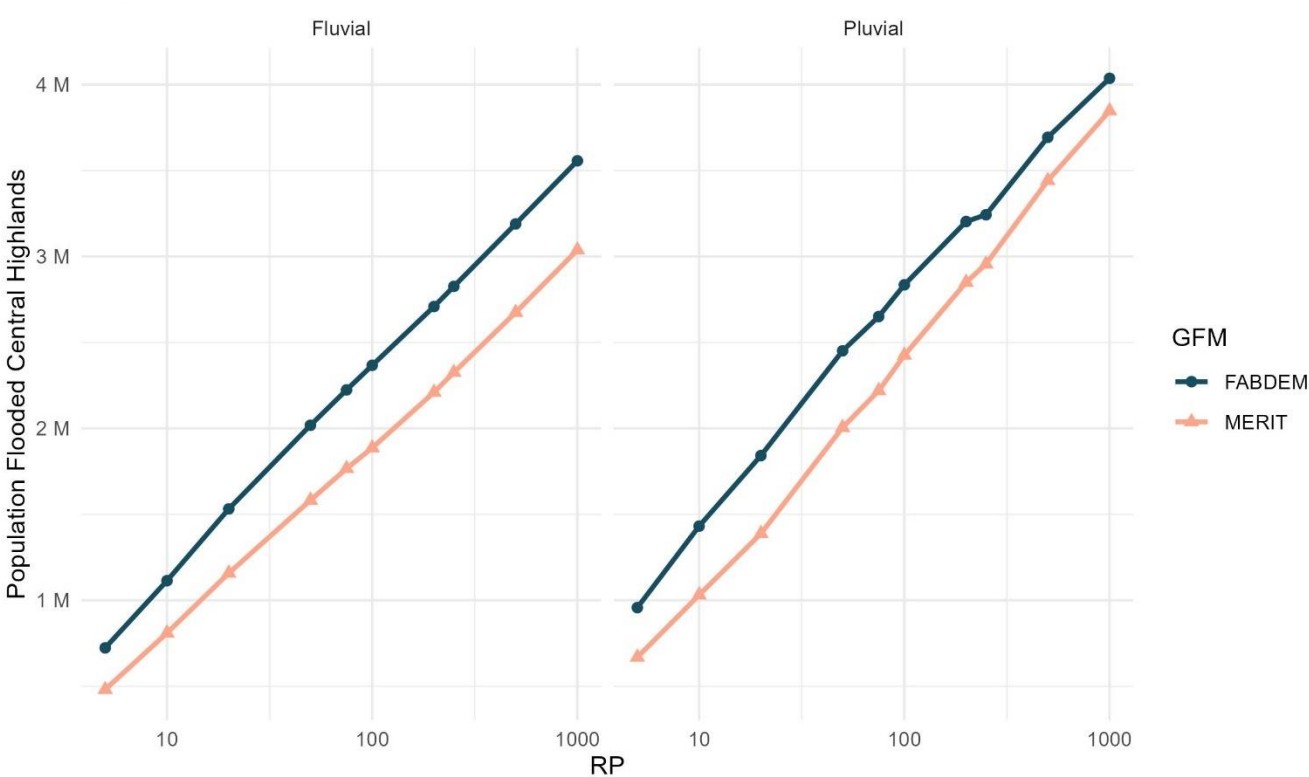


**Figure 13: Total exposure for fluvial and pluvial flooding for the 11 provinces analysed. Flood return periods (RP) range from the 1-in-5 year flood to the 1-in-1000 year flood (x-axis). GFM MERIT model are given in orange and GFM FABDEM in blue.**





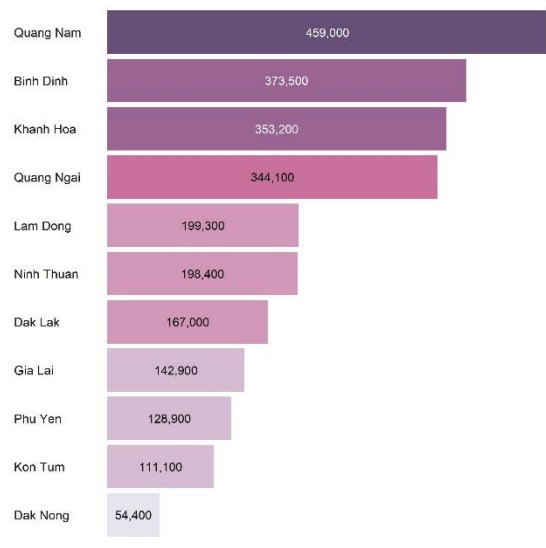

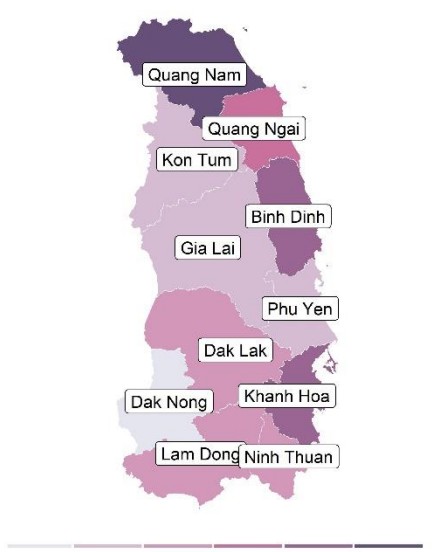

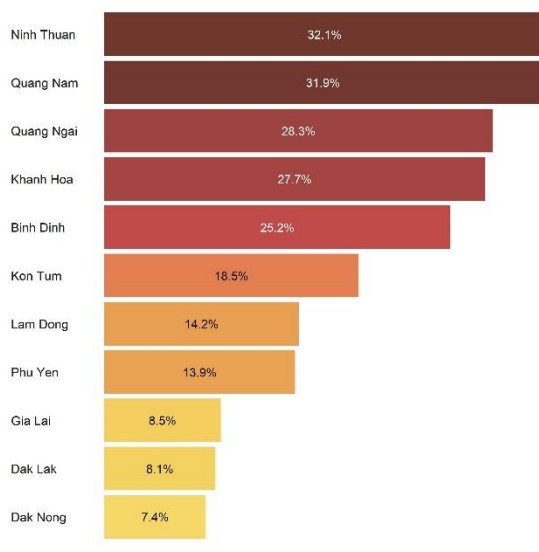

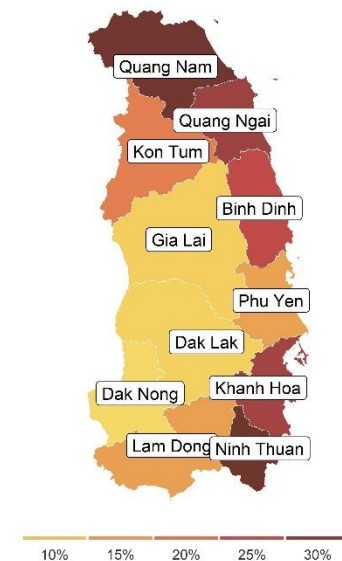

**Figure 14: Flood exposure for the 1-in-100 year flood return period as total numbers per province (purples) and % of total province population (oranges)**




**Table 2: Fluvial flood exposures for the 1-in-20 and 1-in-100 year return period flood for both GFM MERIT and GFM FABDEM**

| Province | GFM FABDEM 1 in 20 | GFM MERIT 1 in 20 | GFM FABDEM 1 in 100 | GFM MERIT 1 in 100 | % Difference GFM FABDEM to GFM MERIT 1 in 20 | % Difference GFM FABDEM to GFM MERIT 1 in 100 |
|---|---|---|---|---|---|---|
| Binh Dinh | 237,300 | 166,600 | 373,500 | 287,300 | 29.80% | 23.10% |
| Dak Lak | 129,400 | 92,700 | 167,000 | 135,400 | 28.40% | 18.90% |
| Dak Nong | 48,200 | 25,800 | 54,400 | 34,400 | 46.50% | 36.80% |
| Gia Lai | 107,400 | 75,900 | 142,900 | 107,300 | 29.30% | 24.90% |
| Khanh Hoa | 242,900 | 185,400 | 353,200 | 283,100 | 23.70% | 19.80% |
| Kon Tum | 97,000 | 71,400 | 111,100 | 97,700 | 26.40% | 12.10% |
| Lam Dong | 150,500 | 80,600 | 199,300 | 132,000 | 46.40% | 33.80% |
| Ninh Thuan | 123,000 | 129,500 | 198,400 | 187,400 | -5.30% | 5.50% |
| Phu Yen | 72,300 | 65,300 | 128,900 | 104,600 | 9.70% | 18.90% |
| Quang Nam | 256,100 | 225,500 | 459,000 | 383,100 | 11.90% | 16.50% |
| Quang Ngai | 182,100 | 120,200 | 344,100 | 263,800 | 34.00% | 23.30% |
| Total | 1,646,200 | 1,238,900 | 2,531,800 | 2,016,100 | 32.80% | 25.50% |







**Table 3: Pluvial flood exposures for the 1-in-20 and 1-in-100 year return period flood for both GFM MERIT and GFM FABDEM**

| Province | GFM FABDEM 1 in 20 | GFM MERIT 1 in 20 | GFM FABDEM 1 in 100 | GFM MERIT 1 in 100 | % Difference GFM FABDEM to GFM MERIT 1 in 20 | % Difference GFM FABDEM to GFM MERIT 1 in 100 |
|---|---|---|---|---|---|---|
| Binh Dinh | 290,500 | 179,300 | 457,800 | 335,600 | 38.3% | 26.7% |
| Dak Lak | 199,400 | 151,100 | 299,700 | 260,700 | 24.2% | 13.0% |
| Dak Nong | 88,300 | 45,800 | 119,500 | 80,500 | 48.1% | 32.6% |
| Gia Lai | 172,700 | 159,300 | 251,100 | 245,500 | 7.8% | 2.2% |
| Khanh Hoa | 217,100 | 160,200 | 375,400 | 291,900 | 26.2% | 22.2% |
| Kon Tum | 99,300 | 110,900 | 137,300 | 156,200 | -11.7% | -13.8% |
| Lam Dong | 217,800 | 141,500 | 293,900 | 233,100 | 35.0% | 20.7% |
| Ninh Thuan | 104,200 | 81,000 | 182,000 | 161,600 | 22.3% | 11.2% |
| Phu Yen | 141,500 | 102,800 | 224,400 | 187,800 | 27.3% | 16.3% |
| Quang Nam | 235,000 | 207,100 | 358,800 | 375,500 | 11.9% | -4.7% |
| Quang Ngai | 207,100 | 156,100 | 320,300 | 271,700 | 24.6% | 15.2% |
| *Total* | *1,972,900* | *1,495,100* | *3,020,200* | *2,600,100* | *32.0%* | *16.2%* |

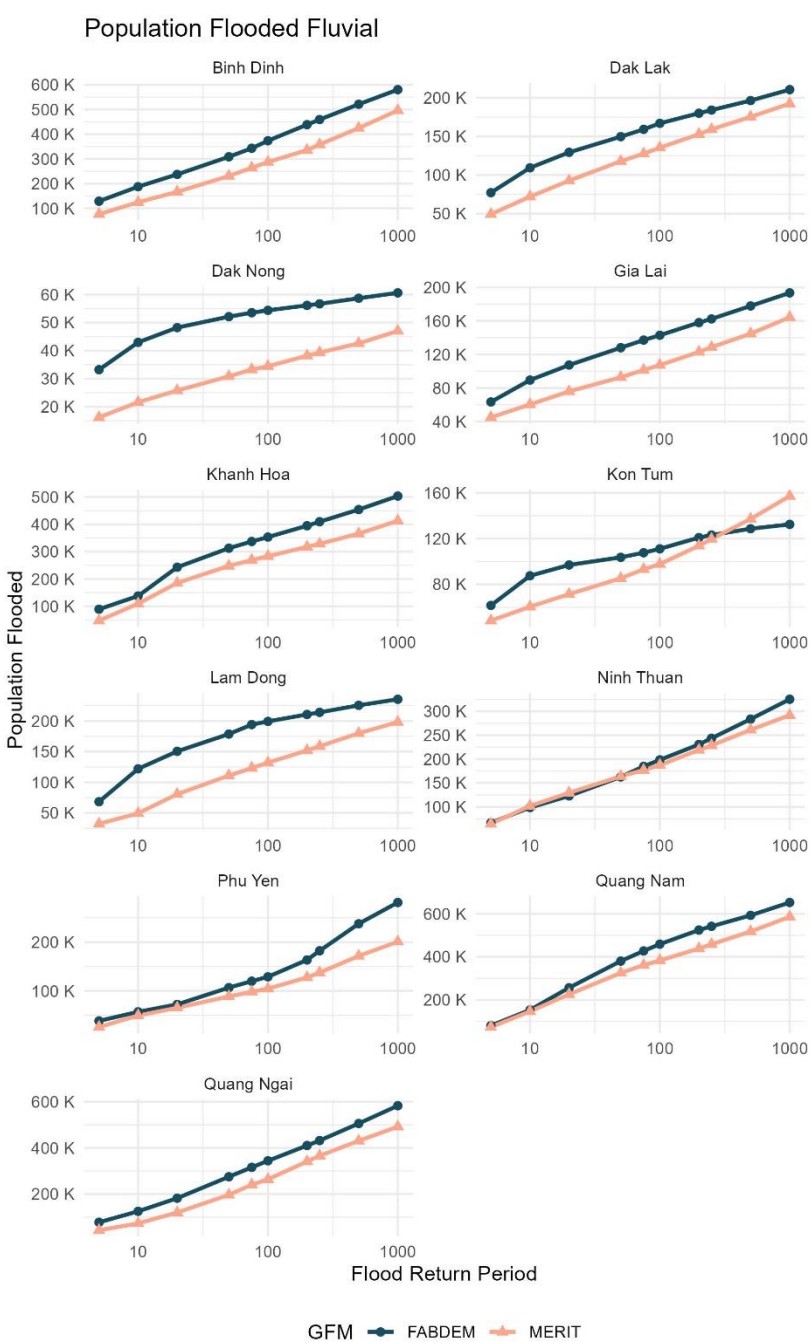

**Figure 15: Flood exposure per province for fluvial flooding. Results shown for GFM MERIT (orange) and GFM FABDEM (blue). Flood return periods (RP) range from the 1-in-5 year flood to the 1-in-1000 year flood (x-axis). Note the y-axis (exposure to flooding) varies per province.**



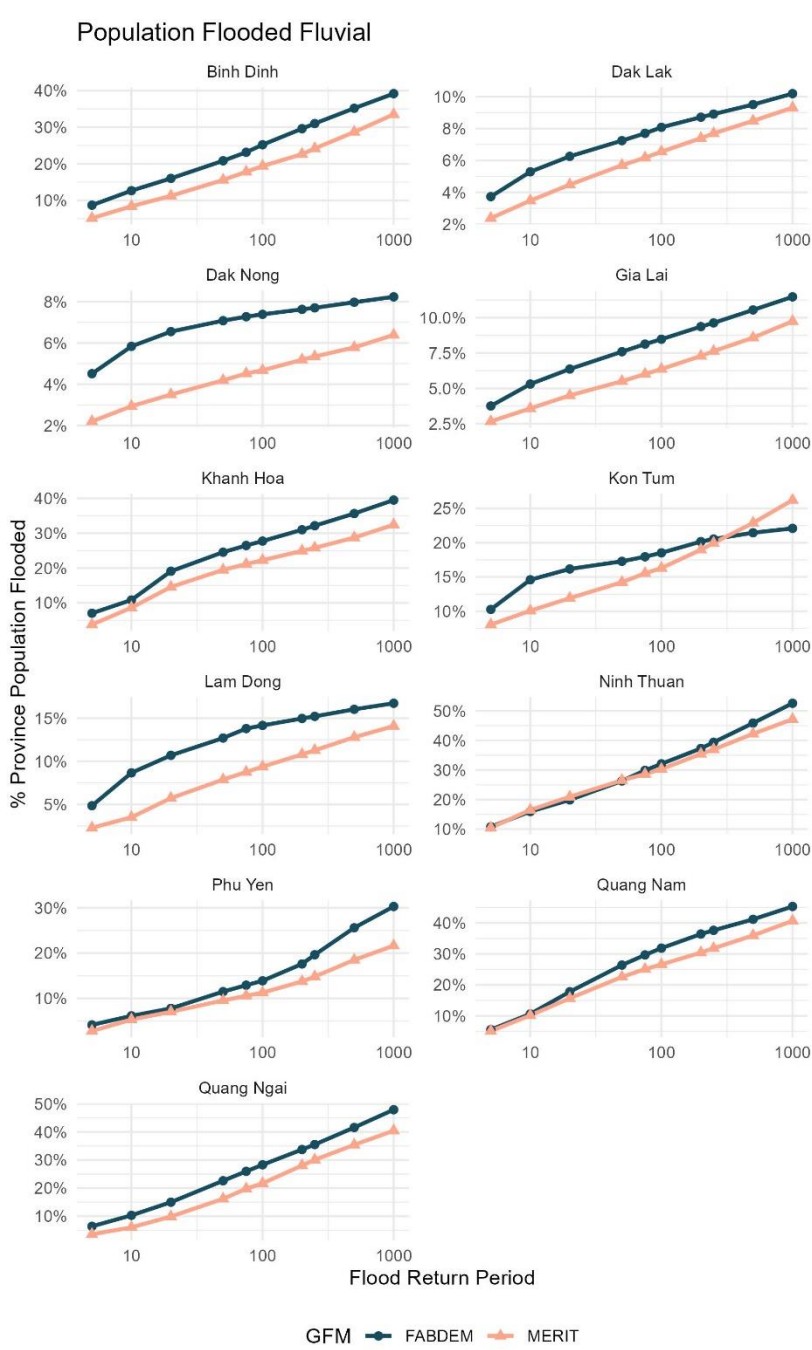

**Figure 16: Flood exposure per province for fluvial flooding as a percentage of total province population. Results shown for GFM MERIT (orange) and GFM FABDEM (blue). Flood return periods (RP) range from the 1-in-5 year flood to the 1-in-1000 year flood (x-axis). Note the y-axis (% province population exposed to flooding) varies per province.**




**Figure 17: Total exposure (y-axis) for fluvial flooding by landcover (rural, peri-urban and urban) for GFM MERIT and GFM FABDEM. Flood return periods (x-axis) range from the 1-in-5 year flood to the 1-in-1000 year flood.**





Figure 18: Total exposure per flood return period as a percentage by landcover (rural, peri-urban and urban) for GFM MERIT and GFM FABDEM (y-axis). The percentages refer to the % of exposure for the associated return period in the given landcover (x-axis).




Figure 19: Total exposure (y-axis) for fluvial flooding by landcover (rural, peri-urban and urban) for GFM FABDEM per province. Flood return periods (x-axis) range from the 1-in-5 year flood to the 1-in-1000 year flood.

**Figure 20: Total exposure per flood return period as a percentage by landcover (rural, peri-urban and urban) for GFM FABDEM (y-axis) per province. The percentages refer to the % of exposure for the associated return period in the given landcover (x-axis).**

## Competing Interests

The authors declare that they have no conflict of interest.





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
