# Peer review of "Assessing LISFLOOD-FP with the next generation Digital Elevation Model FABDEM using household survey and remote sensing data in the Central Highlands of Vietnam"

_Natural Hazards and Earth System Sciences, 2023_

## Referee Comment (RC1)

Review of the manuscript '*Assessing the next generation of Global Flood Models in the Central Highlands of Vietnam*' by *Hawker et al.* submitted to *NHESS*.

Recommendation: ACCEPT

The focus of the paper is on flooding which is a global challenge with impressive damages. The flood impact modelling was introduced and evaluated for several Vietnamese cities / locations through GIS (DEM modelling) and data analysis approaches with aim to determine the risk and expose to flooding in local population divided by peri-urban, urban and rural categories of land use.

Relevance: The presented study is the original primary research within scope of the journal. The manuscript meets general criteria of the significance in hydrological modelling using geoinformation, spatial data (DEM) and GIS tools. The study has been conducted in accordance to the technical standards in mapping and geospatial data analysis and hydrological risk assessment. It is relevant to the journal topic as corresponding to the major domain and research disciplines: Hydrology, nature hazards, *GIS, applied cartography, statistical analysis of spatial data*.

Abstract is well written and clearly describes the undertaken study.

Structure: The article is well organized with structured sections. The structure of the manuscript conforms to the journal standards and discipline norm. It has the following standard sections: 1 Introduction; 2 Hydrodynamic Modelling Materials and methods (subdivided into subsections); 3 Model Evaluation Materials and Methods (ditto); 4 GFM Evaluation (ditto); 5. Flood Exposure for the Central Highlands of Vietnam  (ditto); 6 Conclusions. Sections 2, 3 and 4 are divided into the minor subsections and paragraphs for a better structure and presentation of Methodology. The numeration of the sections is correct and consecutive.

Logic: The clarity of the text logic and organization of the paper is sufficient. It demonstrates the consistent interpretation of the results with detailed explanations of flood modelling approaches and provided comments. A comparison of the results with those in previous studies is presented.

Introduction presents a background regarding the problem of flooding in general and in low and middle-income countries with rapid demographic change in particular. The Introduction defines research goals and provides a clear statement of research problem – to evaluate the impacts of flooding and compare the exposure of local areas using advanced tools of spatial data analysis and modelling. The Introduction well describes the research. Introduction and background show context of the article. Literature is well referenced and relevant.

Study area: is described with sufficient details.

Research questions and goal are identified. Objectives are relevant to the study aim.

Literature regarding the relevant topics is reviewed, formatted according to the journal rules and appropriately referenced. Major sources include published papers on *river flood risk assessment, economic response to floods, evaluation of the disaster risks, general hydrology topics and land cover data management, flood inundation modeling, water resources management*, etc. The referenced literature is within the scope of the study and well cited in the text of the manuscript.

Research gaps and weakness in former works are described – lack of high precision hazard mapping data with which to better understand or manage risk; the existing gaps are identified. The contribution of this work filling this gap is explained.

Motivation is explained: this study contributes to fill in the gaps in the existing similar research through presenting a new approach to the Global Flood Models through developed DEM-based simulation of flood modelling in Vietnam.

English language: fine. The manuscript is written by the UK citizens using professional English language throughout.

Data used in this study are clearly described: The authors used Sentinel 1 synthetic aperture radar imagery which was processed with Google Earth Engine using a change detection. Hydrological data include daily discharge data for Giang Son gauge. The household survey was performed to collect extensive

survey on health, education, occupation, income, consumption, transfers, credit, assets, risk/shocks (947 households and 4,000 individuals). For population grids, they used WorldPop distributes census information from administrative units. For land use type, they used GHS-SMOD data. Everything is well explained with data sources mentioned.

Methods: Methods described with sufficient detail and information. The workflow is well structured and clearly described with sufficient information to reproduce the approach. Yes, mentioned and explained: flood frequency analysis with three model selection criteria: Akaike Information Criterion (AIC), Bayesian Information Criterion (BIC) and the Anderson-Darling goodness-of-fit test (ADC). technically, they used nsRFA package in R. They applied the algorithms of Markov Chain Monte Carlo Simulations (MCMC) using the Bayesian MCMC interface. They evaluated the MERIT and FABDEM based GFM fluvial hazard outputs against remotely sensed flood extents and village level flood data. Then they compared the remote sensing and household survey data and assessed flooded land parcels by target districts. The authors simulated GFM fluvial hazards using data on extreme discharge return periods. The modelling is explained in details. The GFM pluvial hazard simulations was based on FABDEM DEM with provided illustrations comparing observed vs RS-based wet and dry land parcels in the household. Fluvial vs pluvial flood maps are compared for three villages. A series of flood maps is presented in Figs. 7, 8 and 9 with enlarged map fragments.

Results are reported: The authors assessed the Global Flood Model is the Digital Elevation Model (DEM), which represent the terrain without surface objects. They developed and evaluated a next generation of global hydrodynamic flood model based on the recently released FABDEM DEM. They evaluated the model and compared to a previous version using the MERIT DEM at three study sites in the central highlands of Vietnam using two independent validation data sets based on a household survey and remotely sensed observations of recent flooding. The Results are presented with clarity and include description, graphical illustrations, maps and tables with detailed comments.

Discussion interpreted the major outcomes of this study. The discussion is relevant to the initial research goals and well comments the objectives and achieved results. The authors highlighted major achievements of this study and discussed the results of flood modelling. The advantages of the obtained results are described and compared with other existing studies on flood modelling. The Discussion described the issues of methodology and results.

Conclusion Conclusions are well stated, linked to original research question, limited to supporting results and summarized the study with interpretation of facts. The importance of this paper is well summarized as follows: the authors used global flood model based on FABDEM and found that it outperformed a model based on MERIT, and the agreement between the model and remote sensing was greater than the agreement between the two validation data sets. This is useful for similar studies in flood modelling. The conclusions are appropriately stated and connected to the original questions.

Actuality, novelty and importance of the research is clear. It consists in technical approach of flood risk assessment and evaluating the over the region of Vietnam using DEM data, social questionnaire and GIS methods.

Academic contribution: Rigorous investigation performed to a high technical and professional standard in flood modelling and hydrological risk assessment. The paper increases the knowledge in flood risk hazards in Vietnam and in particular, evaluates social expose. The paper combines technical (data modelling), hydrological and GIS approaches which presents a multi-disciplinary study well deserved to be published in *NHESS*.

Figures The authors presented 20 figures (both maps and graphical plots) which are of acceptable quality, easy to read, relevant and suitable. Figures are labelled and appropriately described. They clearly illustrate the results of the undertaken study and the results of flood modelling and ranked areas at risks.

Recommendation: This manuscript can be ==ACCEPTED== based on the detailed report above.

With kind regards,

- Anonymous Reviewer.

19.06.2023.

---

## Author Response (AR1)

Reviewer Response – nhess-2023-93

**Reviewer 1**

Thank you for taking the time to review our manuscript and providing extremely detailed and positive feedback. We have not made any direct changes based on your comments, but have made some changes based on the suggestions of Reviewer 2 (see below).

**Reviewer 2**

We would like to thank the reviewer for spending the time to review our article. The points raised were good, and we have subsequently addressed below.

1. Title and overall

The overall manuscript definitely focused on the update of a DEM part in an existing GFM from MERIT to FABDEM and evaluated its effect on flood simulation performance. "GFM" here is specific to LISFLOO-FP in this study. The title "the next generation of Global Flood Models" may make readers expect broader evolution. Rather, questionnaire survey data is a unique validation as the authors emphasized in the manuscript. I would suggest a more specific title like "Assessing LISFLOOD-FP with the next generation digital elevation model FABDEM using village and remote sensing data in the Central Highlands of Vietnam".

Response: Thank you for your suggestion. We like your title suggestion and have changed the title to like "Assessing LISFLOOD-FP with the next generation Digital Elevation Model FABDEM using household survey and remote sensing data in the Central Highlands of Vietnam"

2. L. 296

Fig. 6(a&b) should be Figs. 7-9 (a&b)? Figure 6 shows the CSI index for FABDEM and MERIT simulations.

Response: Thank you for catching this mistake. This has prompted us to reorder the Figures as the CSI Index Figure was introduced after the maps, but was numbered preceding.

3. L. 369 to 374

Smoother inundation and gradual increase of flood depth in the MERIT GFM is solely based on larger windows in bias removal employed in MERI, or lower spatial resolution also contributes to it?

Response: This is an interesting question and difficult to know for sure. The kernel sizes in pixel terms used in some MERIT bias removal processes were the same as FABDEM, but with the resolution difference the effective distance of the bias removal in FABDEM is smaller.

4. Fig. 6

How can the authors obtain "Village data" and "Remote sensing data" for return periods longer than the ever-largest flood in the past? This is the most important figure to (potentially) support the conclusion that the FABDEM GFM performs over the MERIT GFM; therefore, I strongly recommend to clarify technical processes to plot this figure.

Response: The return period on the x-axis is from the flood model. Thank you for pointing out this ambiguity. We have added this clarification to the Figure caption. Not Fig. 6 is now Fig. 9.

5. L. 377 and Fig. 6

At lower return periods, the FABDEM GFM performed better over the MERIT GFM at Buon Triet, but the opposite at Thon 3. How can the authors conclude that the FABDEM GFM is superior to the MERIT GFM in this figure.

Response: We conclude FABDEM is superior in this case because the performance of the model is higher for a particular magnitude of flood. At Thon 3 MERIT outperformed FABDEM at lower return periods, but this is because the flood extent is substantially less sensitive to flow magnitude in the MERIT model (mainly because of the smoothness of the DEM). Thus it's not a reason to conclude the DEM is more accurate or better from a flood modelling perspective. In our experience of using MERIT DEM in other contexts it's likely that MERIT would struggle not to overestimate a low magnitude flood at Thon 3 when we see such low sensitivity to flow magnitude. In Buon Triet, the tributary to the north of the village is represented by the resolution improvement in GFM FABDEM, and floods at a higher AEP (i.e. more frequent flood) than GFM MERIT.

Other

We have also added an additional reference on Line 185. The study uses FABDEM in a hydrodynamic model. Reference is:

 Iqbal, A., Mondal, M.S., Veerbeek, W., Khan, M.S.A., Hakvoort, H., 2023. Effectiveness of UAV-based DTM and satellite-based DEMs for local-level flood modeling in Jamuna floodplain. Journal of Flood Risk Management.. https://doi.org/10.1111/jfr3.12937